# Improving wheat grain composition for human health by constructing a QTL atlas for essential minerals
Petros P. Sigalas [1,5], Peter R. Shewry [1,5], Andrew Riche[1], Luzie Wingen [2], Cong Feng [3], Ajay Siluveru[2], Noam Chayut[2], Amanda Burridge [4], Cristobal Uauy [2], March Castle[1], Saroj Parmar[1], Charlie Philp[2], David Steele[1], Simon Orford[2], Michelle Leverington-Waite[2], Shifeng Cheng[3], Simon Griffiths [2] & Malcolm J. Hawkesford [1] ✉

Wheat is an important source of minerals for human nutrition and increasing grain mineral content can contribute to reducing mineral deficiencies. Here, we identify QTLs for mineral micronutrients in grain of wheat by determining the contents of six minerals in a total of eleven sample sets of three biparental populations from crosses between A.E. Watkins landraces and cv. Paragon. Twenty-three of the QTLs are mapped in two or more sample sets, with LOD scores above five in at least one set with the increasing alleles for sixteen of the QTLs being present in the landraces and seven in Paragon. Of these QTLs, the number for each mineral varies between three and five and they are located on 14 of the 21 chromosomes, with clusters on chromosomes 5A (four), 6A (three), and 7A (three). The gene content within 5 megabases of DNA on either side of the marker for the QTL with the highest LOD score is determined and the gene responsible for the strongest QTL (chromosome 5A for Ca) identified as an ATPase transporter gene (*TraesCS5A02G543300*) using mutagenesis. The identification of these QTLs, together with associated SNP markers and candidate genes, will facilitate the improvement of grain nutritional quality.

Humans require a range of minerals in their diets, including major elements (calcium, magnesium, phosphorus, sodium, potassium) and trace elements (iron, zinc, fluoride, selenium, copper, chromium, iodine, manganese, molybdenum). However, some of these minerals are also toxic if consumed in excess as are other elements which are normally present in trace amounts such as aluminium, cadmium, mercury, arsenic and lead[1,2].

Most essential minerals are present in adequate amounts in human diets, but deficiencies may occur resulting in severe and widespread symptoms. The most widespread global deficiencies are of iron (Fe) and zinc (Zn). It has been estimated that 43% of children and 29% of women of reproductive age have anaemia, about half of which results from Fe deficiency[3], while Zn deficiency is associated with stunted growth in over 150 million children under the age of 5 globally[4].

The risk of deficiency is reflected in national monitoring of dietary intakes, with the UK National Diet and Nutrition Survey (NDNS) monitoring the intakes of seven minerals: Fe, Zn, calcium (Ca), magnesium

(Mg), potassium (K), selenium (Se) and iodine. In fact, Bates et al. (2016) reported that a substantial proportion of UK children aged 11–18 had intakes below the recommendations for all these minerals, particularly Fe (48% of girls 11 to 18 years)[5]. Similarly, 27% of adult women (19 to 64 years) also had Fe intakes below the recommended level and substantial proportion of adults had intakes below the recommended level for Mg, K and Se.

Wheat is the most widely grown and consumed crop in the world, contributing between 10% and 50% of the total calories in countries ranging from Western Europe to North Africa and Central Asia. Wheat contributes many essential dietary components as well as energy including up to 20% of essential minerals in the UK[6,7]. Hence, deficiencies in the contents of minerals in samples of wheat grain can have significant effects on human health.

The mineral content of wheat grain is determined by the mineral characteristics of the soil and by the ability of the plant to take up minerals from the soil and transport them into the grain. Strategies have therefore

[1]Rothamsted Research, Harpenden, Hertfordshire, AL5 2JQ, UK. [2]John Innes Centre, Norwich, Norfolk, NR4 7UH, UK. [3]Shenzhen Branch, Guangdong Laboratory for Lingnan Modern Agriculture, Genome Analysis Laboratory of the Ministry of Agriculture, Agricultural Genomics Institute at Shenzhen, Chinese Academy of Agricultural Sciences, Shenzhen, 518124, China. [4]School of Biological Sciences, University of Bristol, Bristol, BS8 1UD, UK. [5]These authors contributed equally: Petros P. Sigalas, Peter R. Shewry. ✉e-mail: malcolm.hawkesford@rothamsted.ac.uk

been adopted to increase the mineral contents of wheat grains by either applying minerals as fertiliser (known as agronomic biofortification) or by improving the ability of the plant to extract minerals from the soil and transport them to the grain (known as genetic biofortification). Agronomic biofortification can have a significant impact with some minerals and farming systems. For example, the application of fertiliser containing Se is used in some countries[8], while fertilisation with Zn may also have benefits[9,10]. However, agronomic biofortification adds to costs of crop production and may not be available to farmers in less developed countries.

Hence, genetic biofortification has been the major focus of research globally[11,12]. The contents of Fe and Zn in sets of adapted cultivars have been reported to vary by about 2-fold. For example, Zhao et al. (2009) reported 28.8–50.8 (mean 38.2) ppm Fe and 13.5–34.5 (mean 21.4) ppm Zn in 150 bread wheat genotypes grown in Hungary[13], Morgounov et al. (2007) 25–56 (mean 38) ppm Fe and 20–29 (mean 28) ppm Fe for 66 Central Asian genotypes[14], Joshi et al. (2010) 48.1–50.2 (mean 46.6) ppm Fe and 33.6–34.8 (mean 32.6) ppm Zn for 24 genotypes grown in India[15] and Oury et al. (2006) 27.3–41.9 (mean 34.7) ppm Fe and 16.1–27.2 (mean 20.5) ppm Zn for 51 elite French genotypes[16]. Attempts to exploit this variation have been disappointing, probably due to the combination of multigenic control and strong effect of environment on mineral content. A notable exception is the development of high Zn wheat by CIMMYT. They combined high Zn genes from a number of genetic sources resulting in increases in grain Zn of 30–40% while retaining good agronomic performance[17]. A number of Zn biofortified wheat varieties have been released with grain yields comparable to those of conventional varieties and increases of 8–10 ppm (25–40%) of Zn in the grain[18].

Genetic biofortification depends on the availability of genetic variation in mineral accumulation, either in wheat or in related species which can be used for introgression. It is well-established that modern commercial wheat cultivars are less genetically diverse than older types including landraces (locally adapted types which were grown before the application of scientific breeding methods)[19,20]. We have therefore used the A. E. Watkins landrace cultivar collection to identify quantitative trait loci (QTLs) and genes which determine the accumulation of essential minerals in the grain. The A. E. Watkins collection comprises 826 landrace cultivars which were collected in 32 countries over 90 years ago and has been maintained without selection at the John Innes Centre[21]. Marker analysis showed nine ancestral groups and a core collection of 119 landraces was identified to capture the maximum diversity[21]. The three biparental populations selected for this study were developed from crosses between three landraces of the core A. E. Watkins collection and the spring wheat cv. Paragon[22]. Paragon is a hexaploid UK cultivar which was released in 1998. It was selected after consultation with wheat breeders and researchers as a typical spring wheat genotype for development of genetic resources including EMS and gamma mutant populations (http://www.wgin.org.uk).

In this study, we analysed three biparental populations and determined the content of ten minerals. However, the focus of the current study was on five minerals monitored by the UK NDNS, namely Fe, Zn, Ca, Mg, and K, along with Cu, which may be deficient in livestock diets. This analysis enabled us to identify several QTLs and associated molecular markers for each mineral, which will facilitate the improvement of the nutritional quality of wheat.

## Results

Three populations of recombinant inbred lines (RILs) were selected and grown in replicated multi-environment field trials. The populations were developed from crosses between the UK spring wheat cv. Paragon and Watkins landraces W160, W239 and W292. The landraces originated from Cyprus (W292) and Spain (W160 and W239), representing ancestral groups C7 (W292) and C6 (W160 and W239). The contents of selected minerals and grain yields of the parent lines are shown in relation to the 119 Watkins landraces of the core collection grown in replicated field trials over four years (Supplementary Fig. 1).

Each population comprised 94 F4 RILs which were grown in three replicated randomised plots for three years with either 200 kg N/ha (N2) for Par x W160 and Par x W292 or at two N rates, 200 kg N/ha and 50 kg/ha (N1) in the case of Par x W239. This gave 11 sets of samples which were analysed for nine minerals (Ca, Cu, Fe, K, Mg, manganese, sodium, sulphur and Zn) by ICP-OES. Mineral concentrations were expressed as grain concentration and amount per grain and the discussion below will focus on these primary datasets.

However, a range of other traits were also measured or calculated. The determination of the yields of the plots allowed the total amounts of minerals recovered in grain per square meter to be calculated, referred to as "off-take". Similarly, the relative ability of the lines to accumulate minerals in the grain was calculated as "grain mineral deviation". This is calculated by comparing the concentrations of minerals with the yields of the lines within each set of samples. In broad terms the concentrations of minerals in grain are inversely correlated with grain yield, allowing a regression line to be calculated as discussed for N by Mosleth et al., 2020[23]. Genotypes with mineral concentrations above this regression line exhibit positive grain mineral deviation. Finally, plant height and the weights and concentrations of minerals in straw were also determined. This allowed the weight and mineral content of the above ground biomass and the mineral harvest index to be calculated. Environment and trait abbreviations can be found in Supplementary Data 1. QTL analysis revealed 774 QTLs for minerals in grain, straw, and calculated biomass and 84 QTLs for other traits. Full details of the identified QTLs in Par x W160, Par x W239 and Par x W239 populations can be found in Supplementary Data 2.

Fe, Zn, Ca, Mg and K are of particular interest because they may be deficient in human diets, including developed countries such as the UK. Deficiencies of other essential minerals are rare in humans, particularly in developed countries. However, Cu deficiency may occur in livestock, particularly in cattle[24], hence Cu content is of concern when formulating feeds for livestock. Therefore, the following discussion focuses on the six minerals Fe, Zn, Ca, Mg, K and Cu. The range of concentrations of these minerals and the underlying distributions in the populations are given in Table 1 and Supplementary Fig. 2, respectively (statistical analysis results can be found in Supplementary Data 3).

### Correlations between minerals and between minerals and other traits

Full correlation matrices for all of the traits that were measured or calculated are presented in the Supplementary Data 4, while Fig. 1 shows significant correlations ($p < 0.01$) for each population and N level for mineral concentrations in grain, plant height, straw biomass, above ground biomass, grain yield, harvest index and thousand grain weight (TGW).

Negative correlations between concentrations of some minerals, grain yield and TGW were observed, but these were generally weak, with correlation coefficients below $-0.5$. However, grain Zn concentration showed negative correlations with yield in the majority of environments/populations. This is consistent with the established concept of "yield dilution", as higher yields and larger grain are associated with higher contents of starch which dilutes other grain components. Positive correlations between Zn and Fe concentrations were observed, which were stronger and more consistent in Par x W160, with correlation coefficients ranging from 0.591 to 0.779. Weaker positive correlations between Ca and Mg were observed consistently in populations Par x W239 and Par x W292. For Par x W239, the correlation coefficients ranged from 0.4 to 0.545 for N1 and from 0.287 to 0.5 for N2, while for Par x W292, the correlation coefficients were between 0.458 and 0.655.

### QTLs for essential minerals

A large number of QTLs were identified for mineral concentrations in grain (mg/kg dry weight) and mineral contents per grain (µg/grain). The genome-wide logarithm of odds (LOD) thresholds were calculated based on the permutation test with a significance level (alpha) of 0.05. These thresholds generally fell within the range of 3.0 to 4.0 for various mineral traits.

**Table 1 | Ranges, means, and standard deviations (SD) of grain mineral concentrations in the three populations**

| Population | Year | N Rate | Ca (mg/kg) | | | K (mg/kg) | | | Mg (mg/kg) | | | Fe (mg/kg) | | | Zn (mg/kg) | | | Cu (mg/kg) | | |
|---|---|---|---|---|---|---|---|---|---|---|---|---|---|---|---|---|---|---|---|---|
| | | | Range | Mean | SD | Range | Mean | SD | Range | Mean | SD | Range | Mean | SD | Range | Mean | SD | Range | Mean | SD |
| Par x W160 | 2018 | N2 | 309–515 | 401 | 42.7 | 3618–4719 | 4116 | 257.2 | 915 – 1338 | 1080 | 96.0 | 31.0–46.2 | 38.1 | 3.5 | 24.5 – 43.9 | 33.5 | 3.3 | 4.63 – 6.30 | 5.33 | 0.38 |
| Paragon | | | | 396 | | | 4006 | | | 808 | | | 33.6 | | | 30.0 | | | 5.05 | |
| W160 | | | | 394 | | | 4379 | | | 1268 | | | 38.8 | | | 33.5 | | | 5.43 | |
| Par x W160 | 2019 | N2 | 273–432 | 327 | 29.5 | 3804–4631 | 4165 | 169.3 | 878–1212 | 1027 | 72.4 | 28.3– 46.6 | 35.9 | 3.8 | 23.5–35.5 | 27.9 | 2.2 | 3.63 – 5.42 | 4.53 | 0.36 |
| Paragon | | | | - | | | - | | | - | | | - | | | - | | | - | |
| W160 | | | | 340 | | | 4593 | | | 1260 | | | 50.9 | | | 35.4 | | | 5.13 | |
| Par x W160 | 2020 | N2 | 300–436 | 361 | 32.5 | 3956–4880 | 4424 | 187.7 | 1001 – 1402 | 1185 | 89.4 | 33.7–52.7 | 42.4 | 3.8 | 30.0–43.8 | 35.5 | 3.0 | 4.25–6.05 | 5.09 | 0.38 |
| Paragon | | | | - | | | - | | | - | | | - | | | - | | | - | |
| W160 | | | | 342 | | | 4566 | | | 1361 | | | 41.4 | | | 37.8 | | | 4.84 | |
| Par x W239 | 2015 | N1 | 315–559 | 436 | 52.9 | 3751–5006 | 4251 | 244.9 | 1058 – 472 | 1261 | 83.4 | 34.9–63.3 | 41.9 | 4.8 | 26.5–40.0 | 31.9 | 2.7 | 4.06–5.84 | 4.85 | 0.38 |
| Paragon | | | | 358 | | | 4454 | | | 928 | | | 37.9 | | | 26.5 | | | 4.56 | |
| W239 | | | | 440 | | | 4344 | | | 1532 | | | 48.3 | | | 36.1 | | | 5.01 | |
| Par x W239 | 2016 | N1 | 292–565 | 415 | 43.5 | 3943–5049 | 4421 | 247.4 | 1010 – 1415 | 1209 | 87.1 | 27.7 – 45.9 | 34.5 | 3.5 | 27.5–40.9 | 32.8 | 3.2 | 3.61–5.32 | 4.42 | 0.40 |
| Paragon | | | | 351 | | | 4610 | | | 911 | | | 28.1 | | | 27.0 | | | 4.03 | |
| W239 | | | | 395 | | | 4622 | | | 1446 | | | 44.2 | | | 40.2 | | | 5.14 | |
| Par x W239 | 2016 | N2 | 324– 564 | 431 | 51.2 | 3542–4937 | 4241 | 295.4 | 999 –1337 | 1148 | 77.5 | 31.5–57.1 | 41.1 | 4.5 | 28.4–43.3 | 33.7 | 3.2 | 3.91–5.88 | 4.76 | 0.44 |
| Paragon | | | | 355 | | | 4089 | | | 880 | | | 30.1 | | | 24.8 | | | 4.27 | |
| W239 | | | | 434 | | | 4723 | | | 1335 | | | 47.9 | | | 41.7 | | | 5.18 | |
| Par x W239 | 2017 | N1 | 329–531 | 416 | 45.8 | 3731–4812 | 4322 | 213.2 | 1022–1371 | 1160 | 70.7 | 31.2–79.5 | 44.7 | 9.2 | 24.5–39.2 | 29.6 | 2.6 | 3.97–5.90 | 4.74 | 0.39 |
| Paragon | | | | 369 | | | 4440 | | | 942 | | | 35.1 | | | 27.0 | | | 4.49 | |
| W239 | | | | 407 | | | 4520 | | | 1357 | | | 55.7 | | | 31.1 | | | 4.81 | |
| Par x W239 | 2017 | N2 | 355–632 | 473 | 55.2 | 3568–4555 | 4016 | 201.9 | 1002–1311 | 1127 | 63.3 | 42.2–96.7 | 54.5 | 10.7 | 25.0–42.8 | 34.2 | 2.7 | 3.66–6.35 | 5.18 | 0.42 |
| Paragon | | | | 446 | | | 4179 | | | 988 | | | 58.0 | | | 29.4 | | | 4.98 | |
| W239 | | | | 456 | | | 4092 | | | 1271 | | | 57.0 | | | 38.1 | | | 5.65 | |
| Par x W292 | 2012 | N2 | 380–724 | 509 | 62.4 | 4409–5935 | 5121 | 315.9 | 919–1402 | 1115 | 92.3 | 36.8–64.4 | 47.6 | 5.2 | 28.5–44.6 | 36.0 | 3.2 | 4.71 –7.24 | 5.71 | 0.48 |
| Paragon | | | | 376 | | | 5169 | | | 961 | | | 41.2 | | | 30.4 | | | 5.00 | |
| W292 | | | | 692 | | | 5687 | | | 1453 | | | 55.7 | | | 49.1 | | | 6.95 | |
| Par x W292 | 2013 | N2 | 411–653 | 526 | 62.5 | 3832–5147 | 4296 | 259.5 | 870–1254 | 1067 | 88.3 | 33.5–50.9 | 40.8 | 3.7 | 25.1–38.7 | 31.6 | 2.8 | 4.79–6.92 | 5.74 | 0.42 |
| Paragon | | | | 432 | | | 4387 | | | 875 | | | 35.2 | | | 27.2 | | | 5.14 | |
| W292 | | | | 610 | | | 4559 | | | 1303 | | | 45.9 | | | 37.2 | | | 6.30 | |
| Par x W292 | 2014 | N2 | 386–620 | 475 | 52.5 | 3728– 4636 | 4146 | 203.3 | 834–1248 | 1021 | 73.2 | 33.1–57.0 | 41.9 | 4.5 | 24.4–42.4 | 29.2 | 2.7 | 4.61–6.89 | 5.48 | 0.39 |
| Paragon | | | | 332 | | | 4253 | | | 842 | | | 36.0 | | | 26.0 | | | 4.79 | |
| W292 | | | | 600 | | | 4401 | | | 1240 | | | 59.9 | | | 36.0 | | | 6.51 | |

This table presents the ranges, mean values, and SD of grain mineral concentrations (Ca, K, Mg, Fe, Zn, and Cu) in the three populations across different environments (year, N treatment – where applicable). The mean values (*n* = 3) for the parental lines (Watkins landraces and Paragon) are also provided. *N1* 100 kg/ha, *N2* 200 kg/ha.

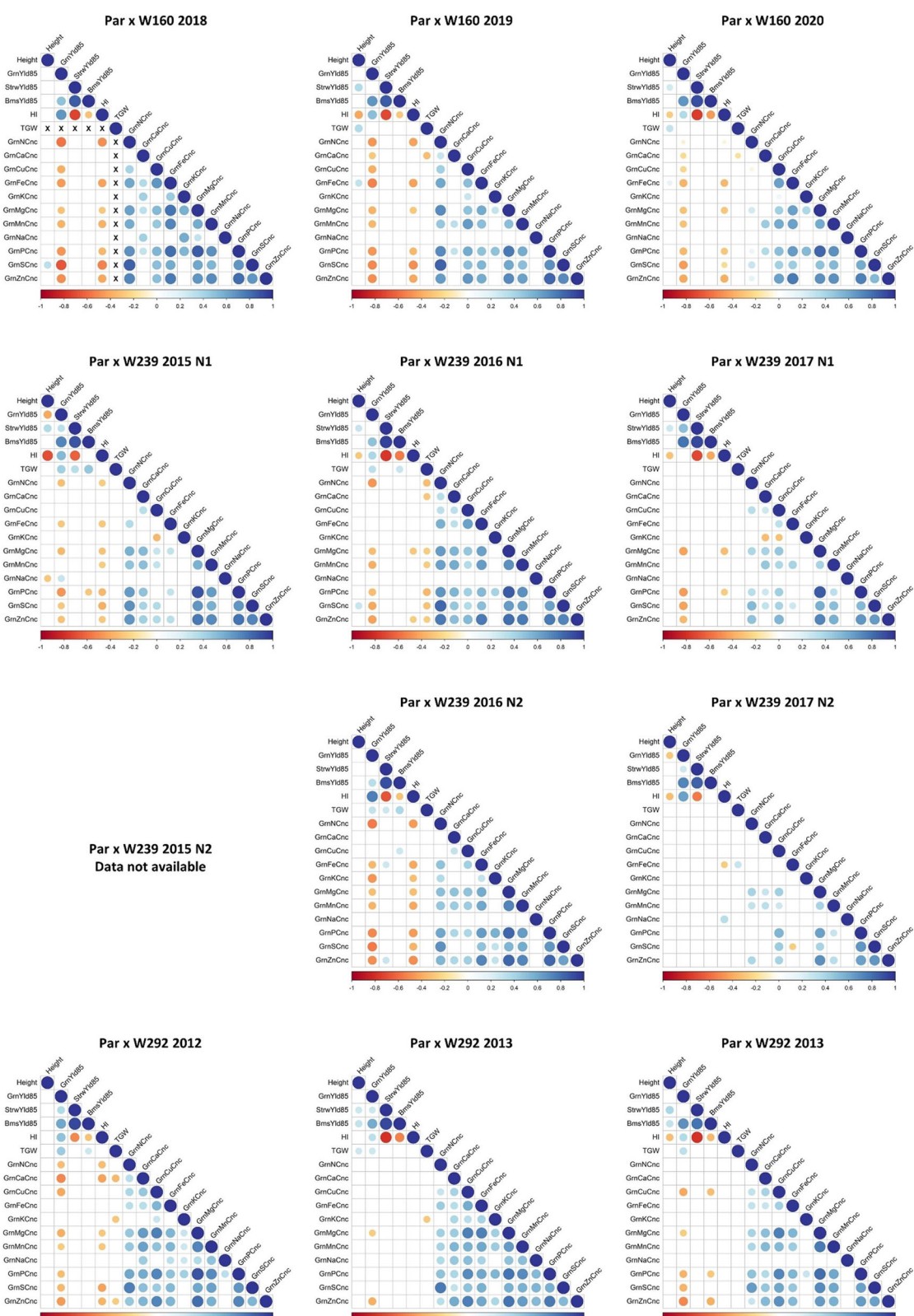

**Fig. 1 | Correlation matrix plots of grain mineral concentration and yield-related traits.** This figure presents correlation matrix plots that illustrate the relationships between grain mineral concentrations and other traits, including plant height, straw biomass, above-ground biomass, grain yield, harvest index, and thousand grain weight (TGW). These correlations are shown for the three biparental populations across different environments (year, N treatment – where applicable). The colour and size of the circles represent the Pearson correlation coefficient. Only correlations that are statistically significant ($p < 0.01$) are depicted. Detailed correlation matrices for all measured or calculated traits are provided in Supplementary Data 4. Environment and trait abbreviations can be found in Supplementary Data 1.

Consequently, it was decided to focus on QTLs that were consistently mapped in at least two sample sets, with LOD scores exceeding 5.0 in at least one set (with one exception which will be discussed below), as these QTLs are more likely of interest for breeding.

Based on these criteria 23 increasing alleles for grain minerals were mapped, with 16 present in the Watkins landraces and seven in Paragon. These are presented in Table 2 which also gives the peak single-nucleotide polymorphism (SNP) markers based on the dataset with the highest LOD score, with additional details presented in Supplementary Data 5. Whereas most of the QTLs were specific for a single mineral, the 7A Cu and 7A Mg

QTLs co-located with each other and with a QTL for grain sulphur concentration (which is listed in Supplementary Data 2 but not discussed here). The number of QTLs for each mineral varied between three (for K and Zn) and five (for Cu) and they were located on 14 of the 21 chromosomes, with three clusters identified on chromosomes 5A (4 QTLs), 6A (3 QTLs) and 7A (3 QTLs) (Fig. 2). The highest LOD score and most consistent QTLs across multiple years for mineral content per grain of each of the essential minerals are presented in Fig. 3.

Some of the QTLs for essential minerals co-located with QTLs for the concentrations/total amounts of the minerals in straw. This is noted in

**Table 2 | QTLs mapped for essential minerals in wheat grain**

| QTL | Trait | Increasing Allele | LOD Scores | Straw QTL Co-location | Peak Marker | Other Minerals | Additive Effect | Variance Explained (%) |
|---|---|---|---|---|---|---|---|---|
| 2B Ca | Ca conc. | P | W292 (5.4), W239 (3.8) | yes | AX-94508946 | | 5.5 | 16.2 |
| 4A Ca | Ca conc. | W | W292 (5.8, 5.2) | no | AX-95630350 | | 23.02 | 19.2 |
| 5A Ca | Ca conc. | W | W239 (12.2, 9.6, 9.5, 9.3, 6.4), W292 (10.2, 9.4, 8.4) | no | AX-86177799 | | 31.81 | 40.4 |
| | Ca/grain | W | W239 (11.7, 10.7, 9.9, 9.5, 6.1), W292 (9.9, 9.8, 8.9) | | AX-643825923 | | 1.34 | 40 |
| 5D Ca | Ca conc. | W | W160 (6.1, 6, 4.7) | no | AX-643802914 | | 16.02 | 30.1 |
| 4B Cu | Cu conc. | P | W239 (8.8, 8.3, 6.7, 5.1) | no | AX-94987158 | | 0.21 | 20.1 |
| 5B Cu | Cu conc. | P | W239 (4.5,3.8, 3.6) | no | AX-94537650 | | 0.12 | 14.6 |
| | Cu/grain | P | W239 (6.7) | | AX-643854728 | | 0.01 | 19.4 |
| 7A Cu | Cu conc. | W | W292 (4.8) | no | AX-94847486 | Mg, S | 0.18 | 18.6 |
| | Cu/grain | W | W292 (5.9, 3.9) | | AX-94847486 | | 0.01 | 23.9 |
| 7B Cu | Cu conc. | W | W239 (3.6, 3.3) | no | AX-94778315 | | 0.14 | 12.6 |
| | Cu/grain | W | W239 (5.6, 3.1) | | AX-94778315 | | 0.01 | 15.6 |
| 7D Cu | Cu conc. | P | W239 (3) | yes | AX-95661674 | | 0.17 | 6.2 |
| | Cu/grain | P | W239 (5.2, 4.3, 3.6) | | AX-643856226 | | 0.01 | 23.5 |
| 2D Fe | Fe conc. | W | W239 (6, 3.4) | yes | AX-94500178 | | 1.62 | 13.2 |
| | Fe/grain | W | W239 (6.3, 5.1, 4.1) | | AX-94500178 | | 0.1 | 27.8 |
| 3A Fe | Fe conc. | W | W239 (5.9), 292 (4.4) | no | AX-643809207 | | 1.3 | 19.1 |
| 5D Fe | Fe conc. | W | W160 (5.5), 292 (4.7) | no | AX-643799519 | | 1.73 | 27.5 |
| 6A Fe | Fe conc. | W | W239 (3.1) | yes | AX-109415640 | | 1.65 | 17.9 |
| | Fe/grain | W | W160 (5.7) | | AX-643821518 | | 0.1 | 22 |
| 3D K | K conc. | P | W239 (5.1, 4.2) | no | AX-95120609 | | 87.17 | 17.4 |
| 4B K | K/grain | W | W239 (8.2, 6.6, 6.4, 5.6) | no | AX-643862089 | | 8.57 | 34.4 |
| 5A K | K conc. | P | W239 (6.3, 4.9) | no | AX-643825923 | | 119.57 | 21.7 |
| 5A Mg | Mg conc. | W | W239 (5, 4.9), W160 (4.1) | no | AX-643824994 | | 28.77 | 15.6 |
| | Mg/grain | W | W239 (4.4) | | AX-643825923 | | 1.83 | 18.6 |
| 6A Mg | Mg conc. | W | W239 (6.4) | no | AX-643809992 | Zn | 44.45 | 18 |
| | Mg/grain | W | W160 (5.4), W239 (4.8) | | AX-643824109 | | 2.12 | 29.8 |
| 6D Mg | Mg conc. | P | W239 (8.3, 4.8, 3.6) | yes | AX-339451161 | | 38.54 | 28.2 |
| 7A Mg | Mg conc. | W | W239 (5), W160 (4.5), W292 (4.1) | no | AX-94535286 | S, Cu | 27.96 | 22.7 |
| | Mg/grain | W | W239 (7.8, 6.6, 3.3), W292 (4.5, 4) | | AX-94535286 | | 2.06 | 20.9 |
| 5A Zn | Zn/grain | W | W292 (5), W239 (3.6) | no | AX-94993840 | | 0.06 | 17.9 |
| 6A Zn | Zn conc. | W | W239 (4.8, 4.7, 4.1) | no | AX-643864818 | Mg | 1.47 | 24.3 |
| | Zn/grain | W | W239 (4.9), W160 (3.7, 3.6) | | AX-643809992 | | 0.1 | 30 |
| 7A Zn | Zn conc. | W | W160 (4.4) | yes | AX-94513729 | | 1.58 | 18.7 |
| | Zn/grain | W | W239 (5), 239 (4.2), 292 (4) | | AX-94863651 | | 0.06 | 15.9 |

This table presents the QTLs for essential minerals including Ca, Cu, Fe, K, Mg, and Zn that have been mapped in at least two sample sets and with LOD scores above 5.0 in at least one set. The table includes information about the trait-increasing allele, LOD scores in each population, co-location with straw QTLs, peak marker, co-location with other minerals, additive effect, and the variance explained by the QTL with the highest LOD score. Additional details can be found in Supplementary Data 2 and 5.

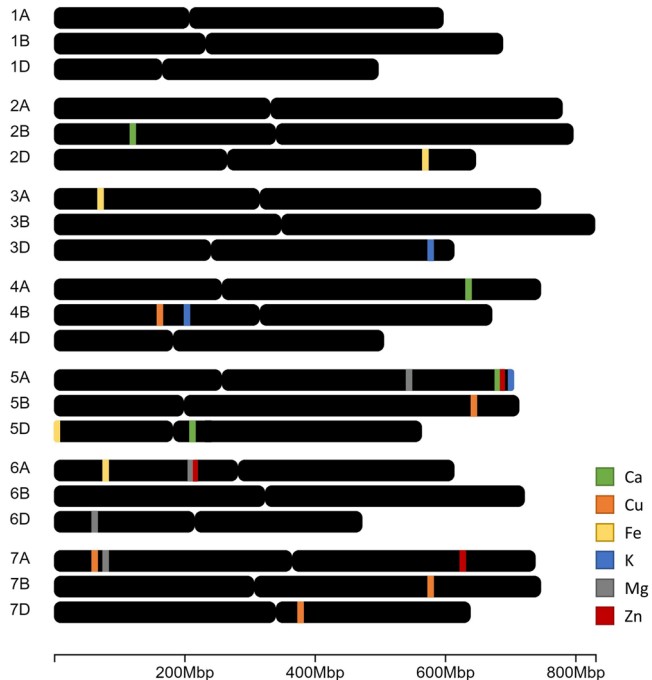

**Fig. 2 | Distribution of identified QTLs for essential mineral content in bread wheat grain.** This figure illustrates the approximate locations of the QTLs across the 21 chromosomes of wheat, based on the IWGSC RefSeq v1.0. Each colour annotation corresponds to a different mineral: Ca (Calcium), Cu (Copper), Fe (Iron), K (Potassium), Mg (Magnesium), and Zn (Zinc), as indicated by the legend. The scale beneath the chromosomes represents distances in Mb. Comprehensive details of the identified QTLs are provided in Table 2 and Supplementary Data 5. Figure was generated using chromoMap v0.4.1.

Table 2 and full details of the QTLs given in Supplementary Data 2. This indicates that the trait is associated with more efficient uptake of the mineral by the plant, whereas the absence of co-located QTLs for minerals in straw/biomass indicates that partitioning of the mineral to the developing grain is more effective.

## Calcium

The grain concentration of Ca in the populations varied from 273 to 1532 mg/kg (Table 1). Increasing alleles for grain Ca were identified in all three Watkins landraces and in Paragon. Increasing alleles were identified on chromosome 5A from both W239 and W292, on chromosomes 5D from W160, and on chromosome 4A from W292. Finally, a Paragon increasing allele was found on chromosome 2B (Table 2).

The 5A Ca QTL had the strongest effect of all of the QTLs mapped in the study and was the most consistent, controlling Ca/grain and Ca concentration in a total of eight sample sets from the two crosses with LOD scores ranging from 6.1 to 12.2 (Table 2). It was therefore selected for proof of function of candidate genes using mutant TILLING lines. Gene content analysis of 5 megabases (Mb) of DNA either side of the marker for the QTL with the highest LOD score revealed the presence of 127 protein-coding genes (listed in Supplementary Data 5). Based on the functional annotation, two candidate genes were identified, *TraesCS5A02G543300* which encodes a cation transporter/plasma membrane ATPase and *TraesCS5A02G542600* which encodes a major Facilitator Superfamily transporter. To further investigate the underpinning gene, mutant lines with non-synonymous mutant alleles in *TraesCS5A02G543300* and *TraesCS5A02G542600* were identified in the bread wheat (cv. Cadenza) TILLING population. In total, eight independent mutant lines were identified for *TraesCS5A02G543300*. Among them, five mutant lines (WCAD1641, WCAD0289, WCAD1253, WCAD1003, WCAD1617) showed a statistically significant increase of more than 10% in Ca grain concentration (Fig. 4, Supplementary Data 6).

Three of those lines showed no change in grain weight, indicating that the observed increase in grain Ca is not simply a result of reduced grain size (Supplementary Data 6). By contrast, none of the mutant lines for *TraesCS5A02G542600* showed a significant change in Ca concentration. These observations therefore indicate that *TraesCS5A02G543300* is the candidate gene under the 5A QTL responsible for the variation in grain Ca content in the two populations.

## Copper

The grain concentration of Cu in the populations ranged from 3.61 to 7.24 mg/kg (Table 1). Five QTLs were identified, with increasing alleles for grain concentration and Cu/grain from W292 on chromosome 7 A and from W239 on chromosome 7B (Table 2). Three QTLs had increasing alleles from Paragon, on chromosomes 4B (Cu concentration), 5D and 7B (both Cu concentration and Cu/grain), with the increasing allele on 4B having LOD scores ranging from 5.1 to 8.8 in four sample sets of the cross with W239.

Two QTLs with high LOD scores and increasing alleles from Paragon were selected to determine gene content: 4B for Cu concentration (LOD 5.1–8.8 in 4 sample sets) and 5B for Cu concentration and Cu/grain (LOD 3.6–6.7), based on 5 Mb of DNA on either side of the marker for the QTL with the highest LOD score. The total numbers of genes identified in the two QTLs were 52 (4B) and 102 (5B), as listed in Supplementary Data 5. Three genes (*TraesCS4B02G131400*, *TraesCS4B02G131500*, *TraesCS4B02G131700*) encoding ZINC-INDUCED FACILITATOR-LIKE protein, and one gene (*TraesCS4B02G128600*) encoding a MULTI-DRUG AND TOXIC COMPOUND EXTRUSION (MATE) protein, were found in the 5 Mb region downstream of the 4B QTL. Notably, genomic comparison between Paragon and W239 of the region surrounding Cu QTL on chromosome 5B revealed the presence of a SNP in the 3'UTR of a gene encoding an ABC transporter C subfamily member (*TraesCS5B02G479900*) present in W239. ABC C subfamily members play a key role in detoxification and metal ion transport.

The identification of Cu QTLs on chromosome 7 of all three subgenomes (7A, 7B, 7D) raises the question of whether the loci are homoeologous. Extraction of the homoeologous regions of each QTL interval from the other two sub-genomes by utilising synteny data and comparison of the chromosome coordinates revealed that the identified QTLs in 7A, 7B, and 7D for Cu do not correspond to homoeologous regions. In total, 101 and 80 protein-encoding genes were found in the 5 Mb region on either side of the 7B and 7D QTL, respectively (Supplementary Data 5). Further analysis revealed that the 7B QTL is located in a high polymorphic region, while some gene deletions were also apparent in W239. Furthermore, gene copy number variations were found in the region around the 7D QTL between Paragon and W239.

## Iron

The grain concentration of Fe in the populations varied by over 2-fold, from about 30 to 64 mg/kg (Table 1). Four QTLs were detected, with increasing alleles from all three Watkins landraces (Table 2). The increasing allele with the highest LOD score was for Fe concentration and Fe/grain on chromosome 2D of W239 (LOD 3.4 to 6.3), with other increasing alleles from W239 and W292 on chromosome 3A, from W160 and W292 on chromosome 5D (both for Fe concentration) and from W239 (Fe concentration) and W160 (Fe/grain) on chromosome 6A.

Analysis of 5 Mb of DNA on either side of the peak marker for the 2D QTL (LOD 6.3) showed the presence of 120 high-confidence protein-coding genes, including transcription factors (Supplementary Data 5). Notably, there is high allelic diversity in this region, with multiple SNPs between Paragon and W239, while large deletions were identified in Paragon. This suggests a divergent genetic background in this locus, possibly a result of introgression events.

## Magnesium

The grain concentration of Mg in the populations varied from 834 to 1532 mg/kg (Table 1). Three QTLs for Mg concentration and Mg/grain

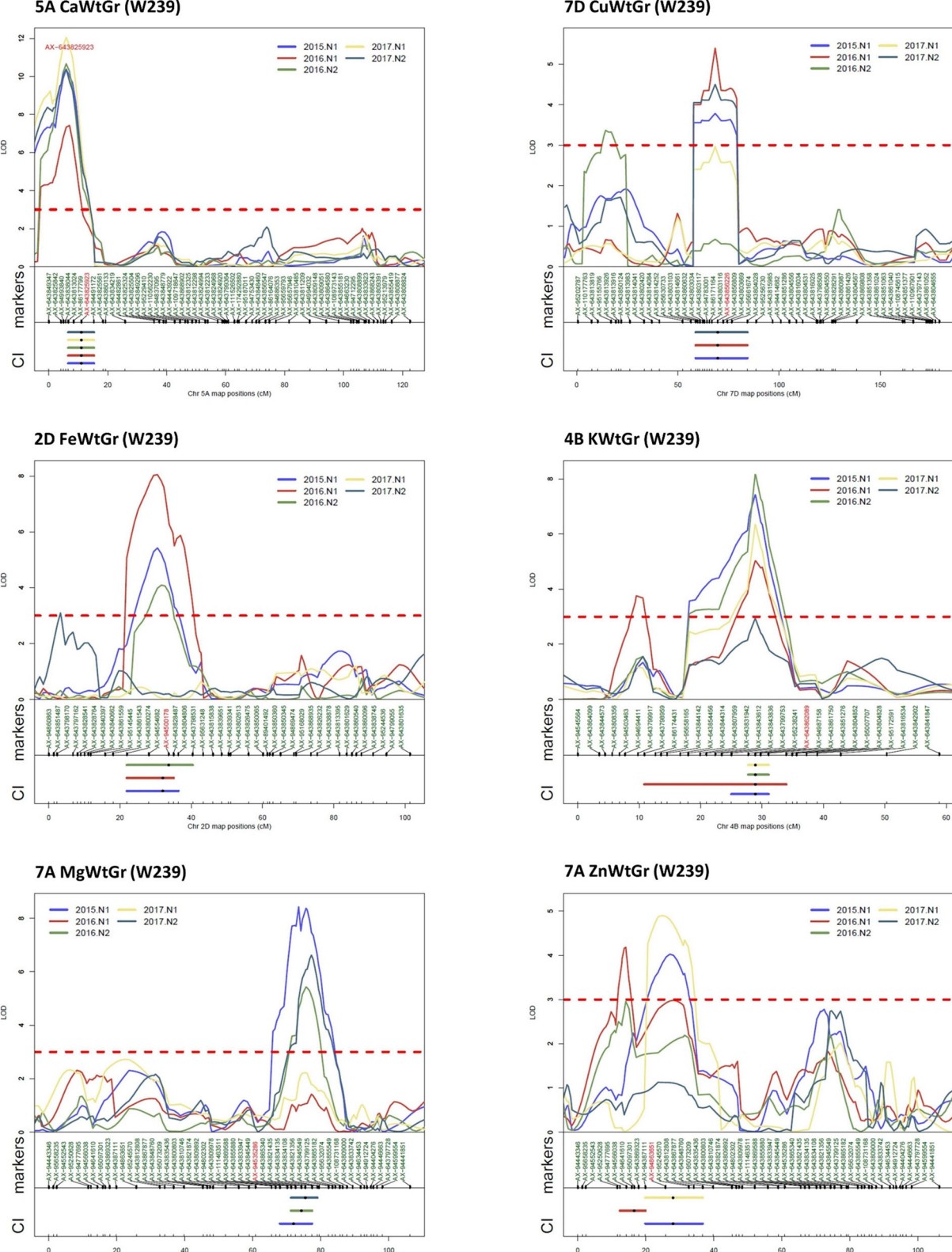

**Fig. 3 | Representative QTL analysis plots.** This figure presents the QTL analysis plots of the highest LOD score and the most consistent QTLs for Ca (5A), Cu (7D), Fe (2D), K (4B), Mg (7A), and Zn (7A) content per grain identified in the Par x W239 population. The vertical axes represent the LOD score, and the horizontal axes correspond to the Axiom35K genetic linkage maps of the respective chromosomes.

Different colours denote different environments (year, N treatment), with the red line indicating a LOD threshold of 3.0. Detailed information about the environments and trait abbreviations can be found in Supplementary Data 1. logarithm of odds (LOD), confidence interval (CI).

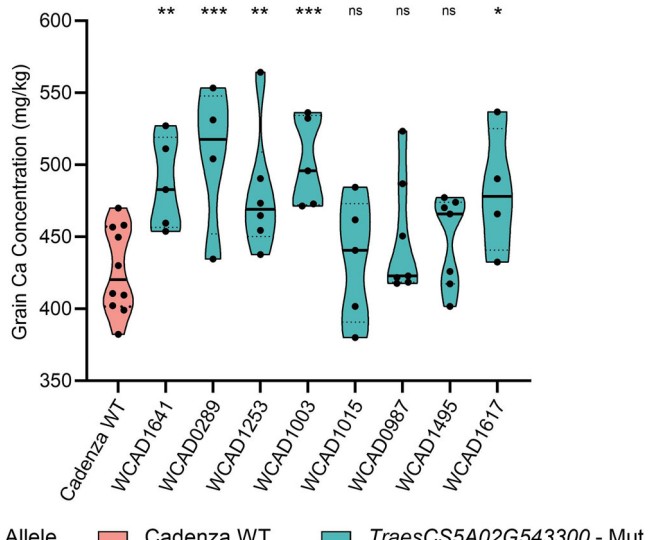

Allele ▉ Cadenza WT   ▉ *TraesCS5A02G543300* - Mut

**Fig. 4 | Violin plots of grain Ca concentration in TILLING mutant lines and WT Cadenza.** This figure presents violin plots of the grain calcium concentration (mg/kg) for eight independent TILLING mutant lines with non-synonymous mutations in *TraesCS5A02G543300* (in blue) and the wild type (WT) Cadenza (in pink). The central solid line in each violin represents the median, and the dashed lines indicate the first and third quartiles. The statistical analysis was conducted using ANOVA. Asterisks indicate statistically significant differences compared to WT Cadenza, with significance levels denoted as follows: * for $p < 0.05$, ** for $p < 0.005$, and *** for $p < 0.0005$. For the WT Cadenza, the sample size (n) is 10 individual plants. For the TILLING mutants, the sample size varies from 4 to 7, as it was limited by the number of homozygous plants detected. Plot was created in GraphPad Prism v9.3.1. Source data can be found in Supplementary Data 6.

were identified with increasing alleles from both W239 and W160 on chromosomes 5A and 6A (Table 2). In addition, a QTL for Mg concentration and Mg/grain was identified on chromosome 7A with increasing alleles in all three Watkins landraces. This was identified in several sample sets with LOD scores for Mg/grain ranging up to 7.8. Finally, a QTL for Mg concentration only was identified in the Par x W239 population with the increasing allele (LOD up to 8.3) from Paragon.

Due to the consistent presence of the 7A Mg QTL across various environments and populations, it was selected for further analysis. The analysis of 5 Mb of DNA surrounding the peak marker for the 7A QTL (with a LOD score of 7.8 in W239) revealed the presence of 128 protein-encoding genes (Supplementary Data 5). There was high genetic heterogeneity across the examined genomic region between Paragon and W239 with many predicted functional SNPs identified in multiple genes. Three stop-gain SNPs were found in the examined region, affecting three genes. The first gene, *TraesCS7A02G126100*, encodes a serine/threonine receptor kinase. The stop-gain SNP in this gene introduces a premature stop codon, potentially leading to a truncated protein with altered functionality. Serine/threonine receptor kinases are crucial components in signal transduction pathways, and any alteration in their structure can impact cellular responses. Stop-gain SNPs were also found in *TraesCS7A02G135200* and *TraesCS7A02G135400*, both encoding MYB-related transcription factors. A stop-loss SNP was found in *TraesCS7A02G130400*, encoding a putative leucine-rich repeat receptor-like protein kinase. Splice region SNPs were identified in seven genes, including those encoding an ATP-dependent zinc metalloprotease (*TraesCS7A02G128400*) and a MATE transporter protein (*TraesCS7A02G131500*).

### Potassium

The grain concentration of K in the populations varied from 3541 to 5935 mg/kg (Table 1). Three QTLs with LOD above 5 were mapped, all in Par x W239 population (Table 2). Two of the increasing alleles for grain

concentration on chromosomes 3D and 5A were from Paragon but the increasing allele for the QTL with the highest LOD scores (from 5.4–8.2 in four sample sets), for K/grain on chromosome 4B, was from W239.

Analysis of the genomic region of the strongest 4B QTL (LOD 8.2) showed that the QTL is in a gene-sparse region as only 29 protein-coding genes were found in the 5 Mb region either side of the peak marker (Supplementary Data 5). Allelic diversity analysis showed the presence of a low number of SNPs, mainly upstream or downstream of the coding region of the genes. However, copy number variation was detected in some genes, indicating structural genomic differences between Paragon and W239 in this locus.

### Zinc

The grain concentration of Zn in the populations varied from 23.6 to 49.1 mg/kg (Table 1). Three QTLs were identified with increasing alleles from the Watkins landraces (Table 2). A QTL for Zn concentration and Zn/grain was identified on chromosome 7A with increasing alleles from all three Watkins landraces, although the LOD scores were low (the highest being 5). Similarly, a QTL for Zn/grain on chromosome 5A had increasing alleles from W292 (LOD 5) and W239 (LOD 3.6). Finally, a QTL for Zn concentration and Zn/grain on chromosome 6A was included although the increasing alleles in W239 had LOD scores of slightly below 5 (ranging from 4.1 to 4.9). An increasing allele for Zn/grain was also mapped in W160 in chromosome 6A with LOD scores of 3.6 and 3.7.

Analysis of 5 Mb of DNA surrounding the peak marker of the 6A QTL (LOD 4.9) showed that the QTL is located in a gene-sparse region containing only 40 protein-coding genes (Supplementary Data 5). Low allelic diversity was found between Paragon and W239, with SNPs mainly located upstream or downstream of the protein-coding regions. Examination of the DNA region extending 5 Mb around the peak marker of the 7A QTL (with a LOD score of 5 in W239) revealed the presence of 91 protein-encoding genes (Supplementary Data 5). Comparative analysis of the genomic region between Paragon and W239 revealed substantial structural variations, characterized by a notable number of SNPs and gene deletions, which might contribute to functional differences between the two genotypes. Predicted functional SNPs were identified in *TraesCS7A02G435500*, encoding a form of calmodulin, which is known to be involved in mineral homeostasis. In addition, a gene encoding a bHLH transcription factor (*TraesCS7A02G435800*) was absent from Paragon.

### Discussion

Wheat is an important source of mineral micronutrients, including minerals that are frequently deficient in human diets. We have therefore exploited genetic variation in wheat landraces and the availability of extensive genomic databases to map QTLs for five minerals which are frequently deficient in UK diets: Fe, Zn, Ca, Mg and K, and for Cu, which may be deficient in livestock diets.

Three recombinant inbred populations were each grown in replicated field trials for three years with one or two levels of nitrogen fertilisation, giving a total of 11 datasets. Furthermore, in order to identify QTLs that could be deployed in high yielding genotypes the grain mineral contents are not only expressed as concentration (as in most published studies) but also as µg/grain. This is important because high concentrations of minerals identified in old types of wheat or wild relatives may be diluted by higher starch accumulation when the trait is introgressed into modern high-yielding germplasm. Hence, the QTLs identified should be robust and amenable to exploitation by wheat breeders.

QTL analysis of the individual sample sets identified a large number of QTLs. Therefore, it was decided to only consider QTLs mapped in at least two sample sets and with a LOD score above 5 in at least one set. In fact, alignment of the QTLs onto the IWGSC RefSeq v1.0 genome assembly (The International Wheat Genome Sequencing Consortium 2018) showed good agreement between the QTLs mapped in the sample sets of each population, and between populations, and LOD scores were often high in several sample sets.

It is of interest that QTLs with increasing alleles from Paragon were identified for Cu, Ca, Mg and K. These four minerals have not been subjected to selection by breeders and our results indicate that there may be sufficient variation in modern elite genotypes for breeders to exploit, rather than requiring introgression of variation from landraces or wild relatives. By contrast, of eight QTLs for Fe and Zn, only one increasing allele was present in Paragon (for Fe) and seven in the Watkins landraces.

The correlations observed between the Fe and Zn concentrations (Fig. 1, Supplementary Data 4) are consistent with previous studies which have reported positive correlations with coefficients up to 0.97[13,14,16,25,26]. It has been reported that multiple ions may share the same transporter, such as Zn, Fe, manganese and cadmium[27]. In fact, we identified some coincident QTLs, for example, for Ca, Zn and K on 5A and for Mg and Zn on 6A. However, it may also reflect broader differences in the efficiency of mineral remobilisation and translocation to the grain during senescence of the vegetative tissues. Irrespective of the mechanism these correlations are relevant to breeding for mineral content and it is notable that the high Zn wheat developed by CIMMYT is also generally high in Fe[28].

Potassium is an essential mineral for humans, particularly as an intracellular electrolyte in the regulation of blood pressure, muscle contraction and nerve transmission. Although dietary K supplies appear to be sufficient at a national level for most countries[29], K deficiency (known as hypokalemia) does occur, including in the UK where it is most prevalent in women[30]. Roots and tubers are the major source of K globally, accounting for up to 80% in some regions, with cereals being the second most important source[29] contributing 15–20% of total K intake in the UK[6,7]. Three QTLs for K were mapped, with the most robust increasing allele (LOD 5.6–8.2 in four sample sets) being from W239.

Deficiencies of Fe and Zn have global impacts on human health (as discussed above). Wheat is an important source of both minerals and mineral enhancement of wheat has therefore been widely studied. The concentrations of Fe and Zn in wheat grain vary depending on the availability of the minerals in soil, with Zn tending to vary more than Fe. The ranges of these minerals in our samples (30 to 64 mg/kg Fe and 23.6 to 49.1 mg/kg Zn) were consistent with studies of multiple genotypes grown on several sites, for example, 28.6–42.5 mg/kg Fe and 20.7–35.2 mg/kg Zn[13], 26.3–49.9 mg/kg Fe and 21.3–64.1 mg/kg Zn[31]. HarvestPlus quote 25 mg/kg as a baseline for Zn with 37 mg/kg as a target, with a bioavailability of 15%. Although HarvestPlus no longer has targets for Fe in wheat, a baseline of 30 mg/kg and target of 59 mg/kg have been reported previously[32].

Numerous QTLs and Marker-Trait Associations (MTAs) for Fe and Zn have been reported, based on Mendelian analysis and Genome Wise Association Studies (GWAS), with at least one QTL and/or MTA being reported on each of the 42 chromosomes[25,26,31,33–37]. Velu et al. (2018) noted that QTLs for "nutrient uptake, transport and sequestration" are clustered on the group 2 and group 7 chromosomes and identified major QTL regions on 2B and 7B[33]. We did not identify QTLs for Fe or Zn on either of these chromosomes. However, QTLs and/or MTAs have been reported on the same chromosomes as those reported here by other studies[25,26,35,37]. It is possible that these corresponded to the QTL reported here but detailed comparisons of the genomic regions are required to confirm this.

Magnesium deficiency in humans has been associated with several adverse health outcomes including cardiovascular disease, hypertension and stroke, metabolic syndrome, type 2 diabetes, Alzheimer's disease and other types of dementia, muscular diseases (muscle pain, chronic fatigue, and fibromyalgia), and types of cancer[38]. Major QTLs for Mg in wheat have not, to our knowledge, been reported previously but Oury et al. (2006) reported wide variation in concentration and high effects of genotype, indicating that it should be amenable to genetic improvement[16]. The four QTLs identified here were all mapped in multiple sample sets and included increasing alleles with high LOD scores from Paragon and Watkins landraces. They therefore provide a good basis for wheat genetic biofortification with Mg.

Calcium deficiency is widespread globally, with up to half of the total population being at risk[39]. Calcium deficiency has a range of adverse health outcomes, including hypertension, high serum cholesterol and increased risk of colorectal cancer, in addition to rickets (paediatric bone disease). Milk and dairy products are the major source of Ca in UK diets (61% of the intake by babies and 35–45% for other age groups), followed by cereals (37% for children aged 11–18)[6,7]. All flours and breads produced in the UK, except wholemeal, are required to be fortified with ≈235–390 mg Ca/100 g flour to restore the level in to that in wholemeal (The Bread and Flour Regulations 1998, legislation.gov.uk). The increasing adoption of vegan diets is a cause of further concern and, Ca intakes from other foods need to be increased[40].

The major QTL for Ca identified on 5A corresponds to a previously identified QTL[41] and analysis of TILLING mutants confirmed the candidate gene as *TraesCS5A02G543300*. This gene encodes a cation transporter/plasma membrane ATPase which is one of seven genes at the QTL which were listed by Alomari et al. (2017)[41]. The 5A Ca QTL had the highest LOD scores of all the QTLs that were mapped in this study, above 10 in some sample sets and controlled both Ca concentration and Ca/grain. In addition, QTLs for Ca were mapped on chromosomes 2B, 4A and 5D. Alomari et al. (2017) also reported a strong marker/trait association for Ca on chromosome 6A[41].

Although Cu is an essential micronutrient for humans it is rarely deficient in human diets. However, Cu deficiency does occur in sheep and cattle, either due to grazing on pastures on low Cu soils (without fertilisation) or due to ingestion of foods high in sulphur and molybdenum[42]. Increasing the content of Cu in feed grain could therefore be advantageous. Our analysis revealed five QTLs for grain Cu, including one on chromosome 4B with a strong increasing allele from Paragon.

The contribution of wheat-based foods to human mineral nutrition is determined by two factors: the concentrations of the minerals in the food and their bioavailability, which are in turn determined by their locations in the grain and chemical forms and other factors.

Fe and Zn are concentrated in the embryo and aleurone layer of the grain, but their relative distributions between these two tissues differ with Fe being more concentrated in the aleurone layer and Zn in the embryo (particularly in the embryonic axis)[28,43]. This results in depleted concentrations of both minerals when grain is milled to produce white flour (the aleurone layer and germ forming part of the bran fraction). For example, Eagling et al. (2014) reported Fe contents of 11.9 mg/kg and 6.7 mg/kg in white flours of two wheat cultivars grown in the UK and 46.7 mg/kg and 30.3 mg/kg in the corresponding whole meal flours[44]. Similarly, Khokhar et al. (2020) reported a range of 24–49 mg/kg of Zn in whole grains of the progeny of crosses between a modern cultivar and land races of bread wheat and 8–15 mg/kg of Zn in white flours of 24 selected genotypes[45].

Furthermore, most of the Fe and Zn in the aleurone cells and in the scutellum of the embryo is present as phytates in discrete bodies known as phytin globoids. Phytates are complexes with phytic acid (inositol hexakisphosphate) which has a cyclic structure with six phosphate groups which can bind metal ions. Phytates have low solubility and hence the bioavailability of Fe and Zn in whole grain wheat is low, although probably higher for Zn (about 25%) than for Fe (about 10%)[46]. The higher bioavailability of Zn could result from the presence of Zn which is not bound to phytin in the embryonic axis. The location of Zn in the genetically biofortified high Zn lines discussed above does not differ from that in conventional lines[28] and consequently the bioavailability may also be limited.

The concentrations of Ca and Mg are much higher than those of Fe and Zn (Table 1). Precise values for Ca and Mg vary between reports but the contents of Mg are generally higher than those of Ca, but with greater proportional losses on milling. For example, Vignola et al. (2016) reported mean contents of about 340 mg/kg Ca in 945 mg/kg Mg in whole grains of 11 wheats grown over two years in Argentina, the values for white flour being 145 mg/kg Ca and 276 mg/kg Mg[47]. Millers quote values of 320 mg/kg Ca and 830 mg/kg Mg in wholemeal and 1340 mg/kg Ca and 260 mg/kg Mg in white breadmaking flours (Nutritional contribution of flour, ukflourmillers.org).

Both Ca and Mg may be bound to phytate in the aleurone layer and scutellum[48] but phytate is not present in the starchy endosperm (the origin of white flour) and hence the minerals should be more bioavailable. It has also

been shown that increasing the fibre content of wheat flour, which is another strategic target for improving health outcomes in western countries, can increase Ca absorption in the human colon, and consequently bone density, probably due to the fermentation to short chain fatty acids that reduce the pH in the colon[49]. Consequently, combining biofortification of wheat flour with Ca and fibre[50] could result in synergistic improvements in health.

## Methods

### Populations
Three biparental segregating populations were developed as described in Wingen et al. (2017) from crosses between the spring bread wheat cv. Paragon as the common variety and a single-seed descendent (SSD) from selected landrace accessions from the A. E. Watkins collection[22]. Details of the selected landraces are given in Supplementary Table 1. Each population comprised 94 F4 recombinant inbred lines. The 35K Axiom Wheat Breeder array was used for population Par x W292 and the 44K Axiom TaNG array for the other two populations and was performed at the Bristol Genomics Facility using established protocols[51].

### Field trials
The field experiments were carried out at Rothamsted Farm, Harpenden, UK (latitude 51.80 N, longitude 0.40 W) between 2012 and 2020. Each population (94 F4 recombinant inbred lines) was grown for three consecutive years, and at one or two levels of N fertilization. The experiments followed a split plot design with blocks split for the N treatment, and with three replicate blocks ($n = 3$). Plot size was 1 ×1 m, and they were sown and harvested with small plot drills and combine harvesters. A seed rate of 350 seeds/m$^2$ was used each year, and all experiments were rainfed with no irrigation. Soil mineral N in the 0–90 cm layer was measured each spring. N was applied in three applications in the form of ammonium sulphate or ammonium nitrate for high N treatments (N2) and once for low N treatments (N1). The first N application each year was as ammonium sulphate. In 2013 the first application was 25 kg of N in the form of ammonium sulphate and 25 kg of N in the form of ammonium nitrate. In 2015, all applications were ammonium nitrate. Fertilizers were applied as solids using standard farm machinery. No fertilizer P or K were applied during the experiments; soil P and K levels are maintained and not considered limiting. Full details of soil type, dates of sowing and harvest and agronomic treatments are given in Supplementary Table 2. Abbreviated environment names are given in Supplementary Data 1. Grain and straw weights per plot were measured on the combine, and sub-samples kept for subsequent analysis.

### Grain and straw analyses
Post harvest, straw and grain samples were analysed for dry matter and mineral content, while the TGW was also recorded. Straw dry matter was determined by weighing before and after drying overnight at 80 °C. Grain dry matter was similarly measured, but with drying at 105 °C. From these data grain and straw yields were corrected for moisture content. TGW was determined by counting 1000 grains, drying at 105 °C overnight and recording the weight. Samples for mineral analysis were oven-dried at 80 °C overnight, weighed and digested using a mixture of nitric acid and perchloric acid (85:15 v/v) in open tube digestion blocks, followed by a programmed heating digestion: 60 °C for 180 min, 100 °C for 60 min, 120 °C for 60 min, 175 °C for 90 min and 50 °C until dry. The acids are removed by volatilisation and the residue dissolved in nitric acid (5% v/v). The elements were quantified with Agilent 5900 SVDV Inductively Coupled Plasma - Optical Emission Spectrometer (ICP-OES) (Agilent Technologies LDA UK Limited, Cheshire, UK). The analysis was strictly monitored using certified external standards alongside in-house standard materials. Standards and check samples are monitored and recorded using Shewhart Control Graphs and computer-based quality control packages.

### Statistics and reproducibility
The impact of genotype on grain mineral concentration was evaluated within each biparental population using one-way analysis of variance (ANOVA) or two-way ANOVA when different N levels were applied. Each population consisted of 94 F4 recombinant inbred lines, each representing a different genotype. For each experiment or year, three biological replicates were included ($n = 3$), with a 1 m x 1 m plot serving as the experimental unit. All statistical comparisons were conducted separately for each year of study using Genstat for Windows 22$^{nd}$ Edition (VSN International, Hemel Hempstead, UK). For the correlation of the different traits, Pearson's correlation coefficients were calculated between the mean values of mineral associated traits, yield related trait and phenotypic data ($n = 3$) using the package *Hmisc v2.0-4*[52] in R software environment *v4.1.1*. Following this, correlation matrix plots were generated using the *corrplot v0.92* package in R[53].

### Quantitative genetics and bioinformatics
The R software environment *v4.3.1* was used for quantitative genetic analysis. Genetic maps were constructed using package *ASMap v1.0-4*[54] following the same strategy as described in Min et al. (2020) using the p-values of 10–14 for Par x W160 and 10–12 for Par x W239 and Par x W292 to define linkage groups[55]. The custom R code to conduct the mapping is deposited at: https://doi.org/10.5281/zenodo.12666456[56]. QTL mapping was conducted using package *qtl v1.50*[57], also as described in Min et al. (2020) using custom made scripts, available from https://doi.org/10.5281/zenodo.12686180[58]. The function 'repeatability' of package *heritability v1.40*[59] was used to calculate the broad sense heritability of the concentrations of minerals in grain and plant height, straw biomass, above ground biomass, grain yield, harvest index and thousand grain weight (Supplementary Data 7). Interesting QTLs were selected using a custom written script, which identified QTL with a LOD over 5, where at least one further QTL on the same chromosome for the same trait and the same effect direction was present. QTLs were aligned along the IWGSC RefSeq *v1.0*, represented by the peak marker, the confidence interval (CI) border markers and all CI internal markers.

### Gene content analysis and genomic comparison
For candidate gene discovery, the analysis was focused on the 5 Mb region either side of the peak marker. In many cases, the confidence intervals of some QTLs were spanning a wide region, making exploring the whole region impractical. The adopted approach might exclude some candidate genes that fall outside this region, but it was empirically selected, allowing a comprehensive search for the most likely candidate genes associated with grain mineral content in bread wheat. The gene ID and genomic location information of the genes within the 5 Mb region either side of the QTL with the highest LOD score for selected traits, as detailed in Supplementary Data 5, were obtained from Ensembl BioMarts[60]. Functional annotation was retrieved from WheatOmics (http://wheatomics.sdau.edu.cn/)[61]. Knetminer was used to explore any association between genes and the traits of interest[62]. Subsequently, whole genome sequence data were used to identify predicted functional SNPs and copy number variations between Paragon and the Watkins landraces as described in Cheng et al. (2024)[63]. Variation data can be accessed from https://opendata.earlham.ac.uk/wheat/.

### Calcium candidate gene proof of function
Wheat lines carrying induced mutations in either of two candidate genes for grain Ca content (GrnCaCnc) *TraesCS5A02G542600* and *TraesCS5A02G543300*, were identified in the Cadenza TILLING population[64] following the method described in ref. 65. Only mutations that were predicted to lead to a gained stop codon, to a missense variant or to a splice donor variant were selected. Two independent mutations were selected for *TraesCS5A02G542600* and eight for *TraesCS5A02G543300*. For each mutation, 24 seeds of the TILLING lines were grown under standard glasshouse conditions. The seeds were placed on wet filter paper and incubated at 48 h at 4 °C and then transferred to 20 °C for germination. After germination (about 3 days) the seedlings were placed to a light peat and sand mixture for seedling development in glasshouse conditions. After rooting the seedlings were transferred to 1 L pots of standard compost mix with

added PG Mix and Osmocote slow-release fertiliser and grown under glasshouse conditions until maturity.

Ten wild-type (WT) Cadenza plants were grown as control. Plants were genotyped with KASP markers specific for the presence/absence of the mutations and only homozygous mutant plants were taken forward (in total 53 plants, between 4 to 7 individual plants for each TILLING mutation). From each of these plants, all grains were harvested and grain number per plant (GNplant), grain yield per plant (GYplant) and the seed characteristics GW, GLng and GWid were measured using a Marvin seed analyser (MARViTECH GmbH, Wittenburg, Germany). Grain moisture content was measured using DA 7250 Near-infrared spectrometer. GrnCaCnc was measured using a benchtop X-ray fluorescence spectrometer, X-Supreme8000, equipped with XSP-Minerals' Package and calibrated with data collected using an ICP-OES using 187 data points with GrnCaCnc levels ranging from 242.4 mg/kg to 726.7 mg/kg. A cross-validation to the calibration was preformed using additional 30 accessions. Linear regression correlation analysis was performed to compare ICP-OES data (ranging from 312 to 460 ppm) with XRF reads. The coefficient of determination, ($R^2$) was 0.823 and Pearson Correlation Coefficient value (R) was 0.907 with $P$ value < 0.001. No outliers for GrnCaCnc were detected and the average over the three technical replicates was calculated. The full dataset, containing grain mineral content and grain characteristics of the TILLING mutant lines, can be found in Supplementary Data 6. This data set (GrnCaCnc range 380.0–564.2 mg/kg, mean 461.3 mg/kg) was used to statistical compare the GrnCaCnc between the WT Cadenza and the independent mutants in a linear model (ANOVA).

## Reporting summary

Further information on research design is available in the Nature Portfolio Reporting Summary linked to this article.

## Germplasm availability

The three biparental populations are all accessible through the John Innes Centre Germplasm Resources Unit (https://www.seedstor.ac.uk/).

## Data availability

Experimental data from field trials and phenotypic information are publicly accessible online at https://grassroots.tools/fieldtrial/all. Variation data are accessible at https://opendata.earlham.ac.uk/wheat/. The results of QTL analysis, correlation analysis, gene content analysis and the source data for the proof of function of candidate genes in the Ca 5A QTL are provided in the paper and its supplementary files, available online. Any further information is available from the corresponding author upon reasonable request.

## Code availability

The custom R code utilized for the genetic mapping and QTL analysis in this study is publicly available at GitHub. The code for genetic mapping can be accessed at https://doi.org/10.5281/zenodo.12666456[56]. Similarly, the code for QTL analysis can be found at https://doi.org/10.5281/zenodo.12686180[58].

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

## Acknowledgements

Rothamsted Research and the John Innes Centre receive strategic funding from the Biotechnology and Biological Sciences Research Council (BBSRC) and we acknowledge support from the Delivering Sustainable Wheat (BB/X011003/1) Institute Strategic Programme. S.C. and C.F. were supported by the National Key Research and Development Program of China (2023YFF1000100), and the National Key R&D Program of China (grant number 2023YFA0914600).

## Author contributions

Conceptualization: M.J.H., S.G., A.R., P.R.S., P.P.S., L.W., S.C., N.C.; Data curation: A.R., L.W., C.F., P.P.S.; Formal analysis: A.R., P.P.S., L.W., C.F., A.B., C.U.; Funding acquisition: M.J.H., S.G., S.C.; Investigation: A.R., S.P., C.F., M.C., D.S., M.L.-W., S.O., A.S., A.B., L.W.; Methodology: A.B.; Project administration: M.J.H., S.G., A.R.; Resources: C.P., N.C.; Supervision: S.G., M.J.H., P.R.S., S.C., N.C.; Validation: A.S.; Visualisation: P.P.S., L.W.; Writing – original draft: P.R.S., M.J.H., S.G., P.P.S., A.R.; Writing – review & editing: P.R.S., M.J.H., S.G., P.P.S., A.R., S.C., L.W., A.B.

## Competing interests

The authors declare no competing interests.
