## [Peer Review File · Communications Biology]

Reviewers' comments:

Reviewer #1 (Remarks to the Author):

1. Brief summary of the manuscript

In this study, QTL mapping for mineral nutrients in wheat grain, straw, and calculated biomass was conducted in three biparental populations from multi-year trials with two added nitrogen levels. TILLING mutants were also examined for two candidate genes contained within a previously identified QTL that was also identified herein for grain calcium concentrations and calcium per grain, and a single candidate gene was identified (one of the seven listed in that previous study).

2. Overall impression of the work

The results section of this manuscript is focused on QTL mapping results for five nutrients (with results for additional nutrients in the supplement). Given that various other studies have examined mineral nutrient concentrations in wheat grain, a more detailed discussion of A) results across multiple tissues and B) main and interaction effects of genotype, year, and nitrogen treatment would be further contributions to the literature and helpful context for the mapping results. Those analysis types appear to be feasible with the data sets presented herein and seem underemphasized in the manuscript as currently written. Additional detail is needed on certain important points within the Materials and Methods as detailed below. Movement of certain text from the Discussion to the Introduction (namely, health implications of certain mineral nutrients) also seems appropriate.

3. Specific comments, with recommendations for addressing each comment

Title: This reviewer would suggest a title that makes it more clear this was an empirical study (rather than a review article).

Abstract (and various places throughout the manuscript): Line 27: How was the search region of ± 5 Mb from a peak marker determined? Similar comment: Lines 227 and 440: Please specify why a ± 5 Mb approach was taken rather than using the confidence interval bounds (e.g., their intersect across sample sets) to narrow down the candidate gene region. Summary statistics on the sizes of the confidence intervals would be helpful to present in the text.

Introduction:

Line 75: It seems a more neutral definition of 'landrace' would be helpful (rather than defining a landrace as being grown before the application of 'scientific' breeding methods). Such definitions appear to commonly involve some cultivation by humans and adaptation to local environmental and management conditions; there is existing literature on this topic, e.g.

<https://www.frontiersin.org/journals/plant-science/articles/10.3389/fpls.2017.00145/full> and <https://www.ncbi.nlm.nih.gov/pmc/articles/PMC3352508/>.

A few points regarding Line 76:

- Slightly more information regarding the Watkins collection in the introduction would be helpful. (E.g., what are the overall size and structure of this collection?)
- It sounds from lines 86-87 as though grain Fe and Zn levels, NUE, and height were examined previously in the three selected landrace parents and other Watkins lines. Where did these three lines reside in the trait distribution within the larger set of Watkins lines (and relative to Paragon itself), for mineral nutrients in particular? (One reason for asking: it sounds like only one of the three Watkins parents was selected due to its grain Fe and Zn levels, but mineral nutrients are the primary focus of this manuscript.) Were the parents included in the field trials as checks? It seems helpful to report their trait values if so, alongside reporting summary statistics of values for each population within each environment-treatment combination.
- A sentence giving more background information on Paragon would be helpful given that it was the common parent; Paragon is currently not mentioned at all in the introduction.
- If space allows, it would be helpful to briefly summarize the ranges of grain mineral nutrient concentrations observed in the literature among modern cultivars, to further justify why biparental populations that each involve a ~100-year old landrace as a parent were examined herein.

Results:

General comment on results section (and lines 322-326): As indicated in the 'overall impression' above, it seems helpful to include a more detailed analysis and description of trait results across years and nitrogen treatments. E.g., 1) a GxExM analysis of mineral nutrient concentrations across years and nitrogen treatments, and 2) an analysis of soil mineral nutrient levels vs. plant mineral nutrient levels, particularly if a main or interaction effect involving year and/or nitrogen treatment was identified for one or more mineral nutrients in one or more plant tissues. The focus of the current results section on mapping results (and one paragraph on correlations, which themselves appear in the supplement) seemed somewhat narrow in scope compared to the analysis types that could be conducted with the data sets presented in this study. Also, the results text describing the mapping results seemed to largely present information that is already concisely conveyed in a table (namely Table 2, if adding the number of genes in the search region as an additional column). It seems that inclusion of one or two additional analysis types for which the collected data sets are suited would be helpful in, and important context for, rounding out the results section.

Line 86: Please state more explicitly why diversity in height was a criterion for selection.

Lines 87-89: This text seems like methods text, as the nitrogen levels are not specified anywhere in the methods. It also seems that planting density and planting dates were not reported, and whether these trials were rainfed or irrigated, etc.

Line 95: Please define 'take-off' and what it is meant to mean when used parenthetically.

Lines 118 and 438 (and various places throughout the manuscript): How was a LOD score of 5 determined as a threshold? It sounds as though permutations were not conducted to establish significance thresholds based on the data sets under examination herein? Similar comment: Line

209: How was a LOD score of 6 determined to be a relevant threshold? It seems helpful to keep a consistent threshold (whether 5, 6, or otherwise).

Lines 127-128: It seems helpful to specify how the 'examples' (or QTL highlighted in Fig. 2) were selected. E.g., solely for traits for which results are described in the main text? LOD scores above a certain threshold? Etc.

Lines 143 and 252: How do these concentrations compare to estimated average requirement in humans and/or HarvestPlus target values for biofortification in wheat?

Line 147: Strongest 'effect' is mentioned, but the only metric reported in this sentence was a LOD score. It seems helpful to either replace the word 'effect' or to add in a metric focused more on effect size to support this sentence. It seems that additive effects are provided in the supplement but not mentioned in the text.

Line 160: Please name the mutants corresponding to TraesCS5A02G542600 for sake of clarity when interpreting Figure 3. Also, it appears 8 mutants (and wild type) are depicted in that figure whereas only 7 mutants (and wild type) are discussed in the text: 5 for TraesCS5A02G543300 and 2 for TraesCS5A02G542600.

Lines 197-8: It would be helpful to describe more clearly to the reader how this check was conducted. E.g., was lack of homeologues determined by checking whether any pairs of genes in the ± 5 Mb regions across subgenomes had greater than a certain percentage of sequence similarity at the protein level?

Line 216: 'linked' how? Please specify. It looks like Wairich et al. was a transcriptomics study? Also, it seems this sentence would be better placed in the discussion rather than the results.

Lines 230, 268, 448, and elsewhere: It seems 'predicted functional' or 'nonsynonymous' (etc.) would be more accurate, as functional validation of the effects of these variants has not been conducted. It is also not clear in line 448 how predicted functional SNPs and copy number variants were identified. At least a brief description of the variant analysis conducted on WGS data would be helpful; it sounds like those details are currently contained in another manuscript that is under review. Similar comment: Line 193: Please clarify how SNPs were found to be 'rare' when only comparing two lines. It sounds like a larger population was likely examined when determining allele frequencies, but this aspect is not described.

Line 246: Does '10 Mb region' here mean ± 5 Mb? It seems helpful to use consistent terminology for this search region, to avoid confusing the reader.

Lines 271-284: It seems helpful to move the correlations subsection before the QTL mapping results, as it is helpful context for those results (e.g., by informing the extent to which each mineral nutrient can be thought of separately vs. anticipating a partially shared genetic and/or environmental basis). It also seems the correlations are sufficiently interesting to be a main figure

(or main table, if not too large), particularly with mineral nutrients having been examined in multiple tissues and agronomic traits also having been examined, which is a strength of this study. Finally, it would be helpful in the discussion to describe how similar your correlation results are with others in the literature (e.g., between mineral nutrient concentrations and thousand-grain weight, as also examined in Krishnappa et al. 2017).

Line 277: Please report correlation coefficients throughout this subsection so that the reader can examine the numerical results alongside the adjectives used to report correlations as weak vs. strong.

Line 282: Please specify the directionality of the correlations between Zn and Fe concentrations.

Discussion:

General comment on discussion: It seems helpful to mention in the discussion whether there are any co-uptake/co-translocation processes (for multiple mineral nutrients) for which to watch out; e.g., Zn and Cd, as reviewed in this paper:
<https://access.onlinelibrary.wiley.com/doi/full/10.2135/cropsci2014.08.0559>

Specific comments on discussion:

Line 307: It is not clear why sufficient variation in ‘modern elite genotypes’ is being discussed when only one such genotype was examined herein. This seems like an overgeneralization unless more context (e.g., from the extant literature) is provided.

Lines 320-321: Zinc-biofortified wheat has been released in six countries by HarvestPlus (https://bpi.harvestplus.org/bpi_cropmaps.html?id=c8). It seems helpful to more fully review the progress made in biofortification of wheat—e.g., relative to target levels for human nutrition—in the introduction (and discussion to contextualize the present results where applicable). Velu et al. (2020), among others, may be relevant to that review:
<https://www.sciencedirect.com/science/article/abs/pii/B9780128184448000055>.

Lines 330-332: Was there any overlap in the present study with the QTL identified in the studies cited in this sentence?

Lines 337-340, 346-353, 362-365, and parallel sentences throughout the discussion for other nutrients: Text regarding health implications of various mineral nutrients (and current fortification schema) seems better placed in the introduction as it is background information for this study.

Lines 353-354: It seems one or more references from the human nutrition realm would be helpful for this sentence.

Line 364: Please lay out more explicitly for the reader why ingestion of foods high in sulphur and molybdenum is relevant for copper deficiency.

Lines 368-369: It seems helpful to add 'among other factors' to the end of this sentence.

Lines 385-387: This recent paper seems potentially relevant in the context of how physical compartments of the grain pertain to mineral nutrient concentrations (and whether biofortification efforts targeting physical traits could be efficacious):

<https://academic.oup.com/g3journal/article/13/7/jkad085/7128031>

Line 394: It seems helpful to switch the order for Ca and Mg when referring to concentrations in white breadmaking flours. (Metrics are reported for Ca and then for Mg throughout the rest of the paragraph, but the order is switched in this last sentence in the paragraph.)

Lines 402-403: Nice ending to the manuscript.

Materials & Methods:

Lines 415-416: Were P and K levels kept constant when altering N levels? It would also be helpful to specify in what form the nitrogen (and fertilization more generally) was applied; formulation, timing and physical method of application, etc.

Lines 420-421: Please report soil mineral nutrient results in the supplement.

Lines 426-7: More information on the protocol used for ICP-AES would be needed, including with reference to a published protocol if applicable. Related comment: Line 90 states 'ICP-OES', but lines 426-7 state ICP-AES; clarification would be needed.

Line 427: Use of the phrase 'a known number' is unclear; were 1,000 grains counted or some subset thereof (and a multiplier then used to reach TGW)? Please specify.

Line 428: Were yield and TGW corrected/standardized for grain moisture? It would likely be helpful to specify either way. It seems this correction/standardization would have been feasible given the grain moisture data available from the NIRS instrument.

Line 437: Which function within the R/qtl package was used for mapping? Were any non-default settings used? Please specify either way.

Line 458: What soil type (and mineral nutrition therein) were used in the glasshouse?

Line 463: Are grain dimension traits (length and width) reported or discussed in this study? They seem potentially relevant, and it was not clear why they are not reported/discussed if they were measured for the TILLING lines and mentioned in the Materials and Methods section.

Lines 464-467: It would be helpful to report performance metrics for the calibration that was developed for the XRF instrument using data from the ICP-OES instrument; e.g., in cross-validation

and on a held-out test set.

Line 486: This reviewer would strongly encourage making the QTL analysis code publicly available at time of publication.

Figures and Tables:

Figure 2: It seems helpful to depict the centromere if not already.

Table 1: It seems mean or median and standard deviation would be easier for readers to compare across populations, years, and nitrogen treatments than ranges.

A few points on Table 2:

- How was 'major' defined? It may be easiest to simply restate the LOD score threshold rather than introducing a new term.
- Adding 'W' in front of 292, 160, and 232 in the fourth column would be helpful in parsing this table.
- If the 'effects on straw and/or biomass' column is indicating whether a QTL was also detected in straw and/or biomass, it seems helpful to make the column header more clear/literal.
- Is 'nearest' marker the same as peak marker? If so, it may be easiest to simply state 'peak marker' rather than introducing a new term.

Reviewer #2 (Remarks to the Author):

Nutritional improvement of major food crops through biofortification is one of the most economical and a food based approach to address micronutrient deficiencies, especially in the developing world. In the present study authors have conducted QTL analyses by phenotyping (minerals, agronomic and yield traits) and genotyping three biparental populations. They detected 774 QTLs and the number of QTLs identified per trait varied. In all 23 robust QTLs were identified, genes were shortlisted for important QTLs.

Over all, experiments have been well carried out and quite a lot of data has been generated. Some of the QTLs/genes identified are good candidates further validation before use in breeding. However, the population size of 94 F4 lines is too small, there is always a chance to identify too many spurious QTLs, use of 5 MB region for candidate genes listing is too big, there is no proper details on genotyping. Also impact of different levels of N on mineral accumulation/QTLs has not been done properly. Use of words such as strong and robust for QTLs needs clear justification or thresholds. Induced mutagenesis for validation of gene for Ca does not make sense because Ca is not major target for biofortification. Experiments have been well carried out and quite a lot of data has been generated but analyses, presentation and interpretation needs a significant improvements. Data can be revisited with the help of biometrician. An indepth and updated analyses looking at the data for multiple aspects can provide a better information for the wheat biofortification. May be a major revision is needed but i leave the final decision to editor

Reviewer #3 (Remarks to the Author):

The manuscript “Improving wheat grain composition for human health: an atlas of QTLs for essential minerals” describes the identification of several QTLs for mineral accumulation in common wheat grain. For one of these QTLs further genetic analyses on EMS mutants led to identify a gene responsible for Ca accumulation on chromosome 5AL.

The study is of interest and it is well written, clear and based on an appropriate experimental design with three RIL populations evaluated in field for three years and at two levels of nitrogen fertilization. I just see no details about statistical analysis, which should be added in order to make the manuscript more complete.

The authors should add, also as supplementary material, the frequency distributions for the traits evaluated in the field trials.

How were the observed means considered statistically different? Based on the phenotypic distribution in the three populations, an ANOVA or a more suitable analysis should be carried out and shown.

Table 1: add heritability and least significant difference to descriptive statistics.

Table 2: Add additive effect and R² for each major QTL. Add also the reference of papers in which the same region has been previously mapped if this is the case.

Response to General Comments

1. Population size of 94 F4 lines was too small.

As demonstrated by Charmez (2000, DOI: 10.1051/agro:2000129), QTL can be detected even in small populations. According to their simulation studies (see Figure 1a – attached below), the empirical power threshold of 90% was exceeded with very small populations ($N = 50$) only for highly heritable QTL ($h^2 > 0.30$). For slightly larger populations ($N = 100$), the threshold was exceeded already with QTL with $h^2 = 0.15$. This population size is very close to the 94 RILs we use in our study. Therefore, while our study may not detect QTL with low heritability, it is still suitable for detecting QTLs with higher heritability, which are of primary interest for breeding purposes.

Figure 1a: Power of QTL detection as a function of population size (N) and QTL heritability: for a centrally located QTL and a dense, regular map. Adopted from: Charmet, Gilles. "Power and accuracy of QTL detection: simulation studies of one-QTL models." *Agronomie* 20.3 (2000): 309-323.

2. Trait results/nitrogen treatments analysis could have been performed.

We understand reviewers' interest in a more detailed analysis of trait results under different N treatments. However, the primary focus of our study is the identification of QTL for improving wheat mineral content, therefore we believe that N-effects are out of the scope of this paper. The inclusion of two different N rates was intended to examine if identified QTLs are consistent under low and high N fertilization rates. Moreover, only one population was grown at two N levels, and data for both levels were collected for only two years, and we tried to clarify this in the manuscript in case it was not clearly mentioned. Given these constraints, we believe that the four sample sets ($2 \times N1$, $2 \times N2$) are not sufficient to carry out a detailed GxExM analysis.

However, we recognize the potential value of such an analysis, therefore, we provide the full datasets for anyone interested in conducting this comparison.

3. The choice of search region of ± 5 Mb from a peak.

We understand this comment about the choice of a ± 5 Mb search region from a peak. In our study, the QTL confidence intervals, which can be found in the supplementary material (Supplementary Table 10 and full detail on Supplementary Tables 3-4), vary significantly depending on the trait and the QTL, ranging from 0 to 170 MB (left) and 0 to 160 MB (right). However, several QTLs showed much smaller confidence intervals of around 10 Mb, similar to the size of our choice of 10 Mb (-5 to +5 Mb either side of the marker).

While using a 5 Mb up- and downstream region is not the perfect approach, we adapted this approach as it allows us to speculate on possible candidate genes without making the analysis too complicated, given that any detected genes will need confirmation in further analysis.

We have added the following in the Materials and Methods section to clarify this: "For candidate gene discovery, we focused our analysis on the 5 Mb region around the peak marker. In many cases, the confidence intervals of some QTLs were spanning a wide region, making exploring the whole region impractical. The adopted approach might exclude some candidate genes that fall outside the region, but it was empirically selected, allowing a comprehensive search for the most likely candidate genes associated with grain mineral content in bread wheat."

Response to Reviewer #1 Comments

- **Title: This reviewer would suggest a title that makes it more clear this was an empirical study (rather than a review article).**

Title modified as follows: Improving wheat grain composition for human health: **constructing** an atlas of QTLs for essential minerals.

- **Line 27: How was the search region of ± 5 Mb from a peak marker determined?**
- **Lines 227 and 440: Please specify why a ± 5 Mb approach was taken rather than using the confidence interval bounds (e.g., their intersect across sample sets) to narrow down the candidate gene region.**

Please see above (response to general comments).

- **Summary statistics on the sizes of the confidence intervals would be helpful to present in the text.**

The confidence intervals varied significantly depending on the trait and the QTL. To avoid a lengthy description in the main text, we have included the confidence intervals for each QTL in the

supplementary materials (Supplementary Table 10). For comprehensive details, readers can refer to Supplementary Tables 3-5. This would allow readers to gain a better understanding of the range and variability of the confidence intervals.

- **Line 75: It seems a more neutral definition of ‘landrace’ would be helpful (rather than defining a landrace as being grown before the application of ‘scientific’ breeding methods). Such definitions appear to commonly involve some cultivation by humans and adaptation to local environmental and management conditions; there is existing literature on this topic, e.g. <https://www.frontiersin.org/journals/plant-science/articles/10.3389/fpls.2017.00145/full> and <https://www.ncbi.nlm.nih.gov/pmc/articles/PMC3352508/>.**

We added “locally adapted types” and included an extra reference

- **Line 76: Slightly more information regarding the Watkins collection in the introduction would be helpful. (E.g., what are the overall size and structure of this collection?)**

We added extra information and reference about A.E Watkins collection: “The A. E. Watkins collection comprises 826 landrace cultivars which were collected in 32 countries over 90 years ago and has been maintained without selection at the John Innes Centre 21. Marker analysis showed nine ancestral groups and a core collection of 119 landraces was identified to capture the maximum diversity”.

- **Lines 86-87: It sounds as grain Fe and Zn levels, NUE, and height were examined previously in the three selected landrace parents and other Watkins lines. Where did these three lines reside in the trait distribution within the larger set of Watkins lines (and relative to Paragon itself), for mineral nutrients in particular? (One reason for asking: it sounds like only one of the three Watkins parents was selected due to its grain Fe and Zn levels, but mineral nutrients are the primary focus of this manuscript.)**

In fact, selection is based on multiyear screening of the core A.E Watkins collection. Data used for the selection of the parental lines were added as **Supplementary Fig. 1**.

The paragraph now reads: “Three populations of recombinant inbred lines (RILs) were selected and grown in replicated multi-environment field trials. The populations were developed from crosses between the UK spring wheat cv. Paragon and Watkins landraces W160, W239 and W292. The landraces were originated from Cyprus (W292) and Spain (W160 and W239), representing ancestral groups C7 (W292) and C6 (W160 and W239). The contents of selected minerals and grain yields of

the parent lines are shown in relation to 119 Watkins landraces grown in replicate field trials over four years (Supplementary Fig. 1).”

- **Were the parents included in the field trials as checks? It seems helpful to report their trait values if so, alongside reporting summary statistics of values for each population within each environment-treatment combination.**

Parental lines of each population (Paragon and the corresponding Watkins landrace) were included in the field trials. Therefore, to address this comment mean values of the parental lines were added in **Table 1**.

- **A sentence giving more background information on Paragon would be helpful given that it was the common parent; Paragon is currently not mentioned at all in the introduction.**

The following was added to address reviewer’ comment: “Paragon is a hexaploid UK cultivar which was released in 1998. It was selected after consultation with wheat breeders and researchers as a typical spring wheat genotype for development of genetic resources including EMS and gamma mutant populations (<http://www.wgin.org.uk>)”.

- **If space allows, it would be helpful to briefly summarize the ranges of grain mineral nutrient concentrations observed in the literature among modern cultivars, to further justify why biparental populations that each involve a ~ 100-year old landrace as a parent were examined herein.**

The following was added in the introduction to address reviewer’s comment: “The contents of Fe and Zn in sets of adapted cultivars have been reported to vary by about 2-fold. For example, Zhao et al. (2009) reported 28.8-50.8 (mean 38.2) ppm Fe and 13.5-34.5 (mean 21.4) ppm Zn in 150 bread wheat genotypes grown in Hungary ¹³, Morgounov et al. (2007) 25-56 (mean 38) ppm Fe and 20-29 (mean 28) ppm Fe for 66 Central Asian genotypes ¹⁴, Joshi et al. (2010) 48.1-50.2 (mean 46.6) ppm Fe and 33.6-34.8 (mean 32.6) ppm Zn for 24 genotypes grown in India ¹⁵ and Oury et al. (2006) 27.3-41.9 (mean 34.7) ppm Fe and 16.1-27.2 (mean 20.5) ppm Zn for 51 elite French genotypes ¹⁶. Attempts to exploit this variation have been disappointing, probably due to the combination of multigenic control and strong effect of environment on mineral content. A notable exception is the development of high Zn wheat by CIMMYT. They combined high Zn genes from a number of genetic sources resulting in increases in grain Zn of 30-40% while retaining good agronomic performance ¹⁷. A number of Zn biofortified wheat varieties have been released with grain yields comparable to those of conventional varieties and increases of 8–10 ppm (25–40%) of Zn in the grain ¹⁸.”

- **General comment on results section (and lines 322-326):** As indicated in the ‘overall impression’ above, it seems helpful to include a more detailed analysis and description of trait results across years and nitrogen treatments. E.g.,
 - 1) a GxExM analysis of mineral nutrient concentrations across years and nitrogen treatments, and
 - 2) an analysis of soil mineral nutrient levels vs. plant mineral nutrient levels, particularly if a main or interaction effect involving year and/or nitrogen treatment was identified for one or more mineral nutrients in one or more plant tissues.

Only one cross was grown at two N levels (W219) with data for both levels for only two years. We do not think that these four (2 x N1, 2 x N2) samples sets are sufficient to carry out a detailed GxExM analysis. However, we provide the full datasets if the reader wishes to carry out this comparison. Please refer also to our response to general comments.

- **The focus of the current results section on mapping results (and one paragraph on correlations, which themselves appear in the supplement) seemed somewhat narrow in scope compared to the analysis types that could be conducted with the data sets presented in this study.**
- **Also, the results text describing the mapping results seemed to largely present information that is already concisely conveyed in a table (namely Table 2, if adding the number of genes in the search region as an additional column). It seems that inclusion of one or two additional analysis types for which the collected data sets are suited would be helpful in, and important context for, rounding out the results section.**

We have simplified the description. We have not included additional data analyses as we wish to focus solely on the key nutritional traits rather than physiological observations.

- **Line 86: Please state more explicitly why diversity in height was a criterion for selection.**

Height, which is indicative of biomass accumulation, can influence mineral content. Consequently, it was one of the traits for which the parental lines were selected. Line 86 has now been amended.
- **Lines 87-89: This text seems like methods text, as the nitrogen levels are not specified anywhere in the methods. It also seems that planting density and planting dates were not reported, and whether these trials were rainfed or irrigated, etc.**

We consider that this short description is helpful for the reader as it provides an overview without detailed reading of the methods (at the end of the paper). We have therefore retained it but revised as follows: “Each population comprised 94 F4 RILs which were grown in three replicated

randomised plots for three years with either 200 kg N/ha (N2) for Par x W160 and Par x W292 or at two N rates, 200kg N/ha and 50 kg/ha (N1) in the case of Par x W239.”

We have also revised the corresponding Materials and Methods section, Field Trials (Line 499-514), and added a supplementary table giving details of the site, agronomy and relevant dates (Supplementary Table 22)

- **Line 95: Please define ‘take-off’ and what it is meant to mean when used parenthetically.**

We edited this in the text to “off-take” (as used in Supplementary Table 2). We amended the sentence to indicate the definition more clearly: “The determination of the yields of the plots allowed the total amounts of minerals recovered in grain per square metre to be calculated, referred to as “off-take.”

- **Lines 118 and 438 (and various places throughout the manuscript): How was a LOD score of 5 determined as a threshold? It sounds as though permutations were not conducted to establish significance thresholds based on the data sets under examination herein?**

All detected QTL were significant. This was determined by performing a permutation test to get a genome wide LOD significance threshold (significant level $\alpha = 0.05$). The determined LOD thresholds for the different mineral traits were in general between 3.0 and 4.0 and always under 5.0. We use the threshold LOD 5.0 to select stronger QTL as they are more likely of interest for breeding.

- **Similar comment: Line 209: How was a LOD score of 6 determined to be a relevant threshold? It seems helpful to keep a consistent threshold (whether 5, 6, or otherwise).**

This is a mistake and should be 5. We have removed the sentence “However, the 3A, 5D and 6A alleles had LOD scores below 6.” which was a casual description. Threshold was decided as described in the previous comment.

- **Lines 127-128: It seems helpful to specify how the ‘examples’ (or QTL highlighted in Fig. 2) were selected. E.g., solely for traits for which results are described in the main text? LOD scores above a certain threshold?**

To address this comment the above was revised as follows:

“The highest LOD score and most consistent QTLs across multiple years for mineral content per grain of each of the essential minerals are presented in Fig. 3.”

“Fig. 3: Representative QTL Analysis Plots - This figure presents the QTL analysis plots of the highest LOD score and the most consistent QTLs for Ca (5A), Cu (7D), Fe (2D), K (4B), Mg (7A), and Zn (7A) content per grain identified in the Par x W239 population. The vertical axes represent the LOD score, and the horizontal axes correspond to the Axiom35K genetic linkage maps of the respective chromosomes. Different colours denote different environments (year, N treatment), with the red line indicating a LOD threshold of 3. Detailed information about the environments and trait abbreviations can be found in Supplementary Tables 1 and 2.”

- **Lines 143 and 252: How do these concentrations compare to estimated average requirement in humans and/or HarvestPlus target values for biofortification in wheat?**

We have added the following to the discussion:

“HarvestPlus quote 25 mg/kg as a baseline for Zn with 37 mg/kg as a target, with a bioavailability of 15%. HarvestPlus no longer has targets for Fe in wheat but has previously reported a baseline of 30 mg/kg and target of 59 mg/kg ³².”

- **Line 147: Strongest ‘effect’ is mentioned, but the only metric reported in this sentence was a LOD score. It seems helpful to either replace the word ‘effect’ or to add in a metric focused more on effect size to support this sentence. It seems that additive effects are provided in the supplement but not mentioned in the text.**

Changed “strong” or “robust” QTL throughout the manuscript to “highest LOD score”. In addition, additive effect and variance explained by each QTL were also added in Table 2.

- **Line 160: Please name the mutants corresponding to TraesCS5A02G542600 for sake of clarity when interpreting Figure 3. Also, it appears 8 mutants (and wild type) are depicted in that figure whereas only 7 mutants (and wild type) are discussed in the text: 5 for TraesCS5A02G543300 and 2 for TraesCS5A02G542600.**

To address this comment, we changed this part in the manuscript to make it more clear, as follows:

“To further investigate the underpinning gene, mutant lines with non-synonymous mutant alleles in TraesCS5A02G543300 and TraesCS5A02G542600 were identified in the bread wheat (cv. Cadenza) TILLING population. In total, eight independent mutant lines were identified for TraesCS5A02G543300. Among them, five mutant lines (WCAD1641, WCAD0289, WCAD1253, WCAD1003, WCAD1617) showed a statistically significant increase of more than 10% in Ca grain concentration (Fig. 4). Three of those lines showed no change in grain weight, indicating that the observed increase in grain Ca is not simply a result of reduced grain size. By contrast, none of the

mutant lines for TraesCS5A02G542600 showed a significant change in Ca concentration. These observations therefore indicate that TraesCS5A02G543300 is the candidate gene under the 5A QTL responsible for the variation in grain Ca content in the two populations.”

- **Lines 197-8: It would be helpful to describe more clearly to the reader how this check was conducted. E.g., was lack of homeologues determined by checking whether any pairs of genes in the ± 5 Mb regions across subgenomes had greater than a certain percentage of sequence similarity at the protein level?**

In response to this comment, this was not conducted by comparing protein sequence similarity. Instead, our approach was based on examining the synteny between sub-genomic regions. Specifically, we extracted the homoeologous regions corresponding to each QTL interval from the other two sub-genomes. We then compared the chromosome coordinates of these regions. Our analysis revealed no overlap between the coordinates, thereby confirming that the identified QTLs in 7A, 7B, and 7D for Cu do not correspond to homoeologous regions.

We therefore modified the text to make this clearer: “Extraction of the homoeologous regions of each QTL interval from the other two sub-genomes by utilising synteny data and comparison of the chromosome coordinates revealed that the identified QTLs in 7A, 7B, and 7D for Cu do not correspond to homoeologous regions.

- **Lines 230, 268, 448, and elsewhere: It seems ‘predicted functional’ or ‘nonsynonymous’ (etc.) would be more accurate, as functional validation of the effects of these variants has not been conducted.**

As suggested by the reviewer this was changed to “predicted functional SNPs” throughout the manuscript.

- **It is also not clear in line 448 how predicted functional SNPs and copy number variants were identified. At least a brief description of the variant analysis conducted on WGS data would be helpful; it sounds like those details are currently contained in another manuscript that is under review.**

For more details, the code etc please refer to [Cheng, S. et al. Harnessing Landrace Diversity Empowers Wheat Breeding for Climate Resilience. bioRxiv \(2023\) doi:10.1101/2023.10.04.560903](https://doi.org/10.1101/2023.10.04.560903). Full genome sequencing data used are also publicly available for anyone interested to repeat this analysis.

- **Similar comment: Line 193: Please clarify how SNPs were found to be ‘rare’ when only comparing two lines. It sounds like a larger population was likely examined when determining allele frequencies, but this aspect is not described.**

The sentence in question referred to the presence of the SNP in the full A.E Watkins landrace collection. However, we agree that this comparison may not be directly relevant to the focus of the current paper and could potentially cause confusion. Therefore, we have decided to remove this sentence to maintain clarity.

- **Line 246: Does ‘10 Mb region’ here mean ± 5 Mb? It seems helpful to use consistent terminology for this search region, to avoid confusing the reader.**

This was now changed and reads as follows: “Analysis of the genomic region of the strongest 4B QTL (LOD 8.2) showed that the QTL is in a gene-sparse region as only 29 protein-coding genes were found in the 5 Mb region either side of the peak marker (Supplementary Table 18).”

- **Lines 271-284: It seems helpful to move the correlations subsection before the QTL mapping results, as it is helpful context for those results (e.g., by informing the extent to which each mineral nutrient can be thought of separately vs. anticipating a partially shared genetic and/or environmental basis).**

In response to this suggestion, we have restructured the manuscript and moved the correlations subsection ahead of the QTL mapping results.

- **It also seems the correlations are sufficiently interesting to be a main figure (or main table, if not too large), particularly with mineral nutrients having been examined in multiple tissues and agronomic traits also having been examined, which is a strength of this study.**

We have moved the correlation matrices between the different traits in each of the sample sets from the supplementary section to the main body of the paper as Figure 1.

- **Finally, it would be helpful in the discussion to describe how similar your correlation results are with others in the literature (e.g., between mineral nutrient concentrations and thousand-grain weight, as also examined in Krishnappa et al. 2017).**

The following has been added to the discussion:

“The correlations observed between the Fe and Zn concentration (Fig. 1, Supplementary Tables 6-9) are consistent with previous studies which have reported positive correlations with coefficients up to 0.97^{13,14,16,26,27}. It has been reported that multiple ions may share the same transporter, such

as Zn, Fe, Mn and Cd ²⁸. In fact, we identified some coincident QTLs, for example, for Ca, Zn and K on 5A and for Mg and Zn on 6A. However, it may also reflect broader differences in the efficiency of mineral remobilisation and translocation to the grain during senescence of the vegetative tissues. Irrespective of the mechanism these correlations are relevant to breeding for mineral content and it is notable that the high Zn wheat developed by CIMMYT is also generally high in Fe ²⁹.”

- **Line 277: Please report correlation coefficients throughout this subsection so that the reader can examine the numerical results alongside the adjectives used to report correlations as weak vs. strong.**

We have now modified the text to include some the correlation coefficients.

- **Line 282: Please specify the directionality of the correlations between Zn and Fe concentrations.**

This changed as follows: “Positive correlations between Zn and Fe concentrations were observed,”.

- **General comment on discussion: It seems helpful to mention in the discussion whether there are any co-uptake/co-translocation processes (for multiple mineral nutrients) for which to watch out, e.g., Zn and Cd, as reviewed in this paper:**
<https://access.onlinelibrary.wiley.com/doi/full/10.2135/cropsci2014.08.0559>

“The correlations observed between the Fe and Zn concentration (Fig. 1, Supplementary Tables 6-9) are consistent with previous studies which have reported positive correlations with coefficients up to 0.97 ^{13,14,16,26,27}. It has been reported that multiple ions may share the same transporter, such as Zn, Fe, Mn and Cd ²⁸. In fact, we identified some coincident QTLs, for example, for Ca, Zn and K on 5A and for Mg and Zn on 6A. However, it may also reflect broader differences in the efficiency of mineral remobilisation and translocation to the grain during senescence of the vegetative tissues. Irrespective of the mechanism these correlations are relevant to breeding for mineral content and it is notable that the high Zn wheat developed by CIMMYT is also generally high in Fe ²⁹.”

- **Line 307: It is not clear why sufficient variation in ‘modern elite genotypes’ is being discussed when only one such genotype was examined herein. This seems like an overgeneralization unless more context (e.g., from the extant literature) is provided.**

Variation in modern genotypes is now included in the introduction (see comment above).

- **Lines 320-321: Despite this global interest and massive investment, including the HarvestPlus programme in CGIAR institutes (<https://www.harvestplus.org>), progress has been limited.**

- **Zinc-biofortified wheat has been released in six countries by HarvestPlus (https://bpi.harvestplus.org/bpi_cropmaps.html?id=c8). It seems helpful to more fully review the progress made in biofortification of wheat—e.g., relative to target levels for human nutrition—in the introduction (and discussion to contextualize the present results where applicable). Velu et al. (2020), among others, may be relevant to that review: <https://www.sciencedirect.com/science/article/abs/pii/B9780128184448000055>.**

The following was added: “A notable exception is the development of high Zn wheat by CIMMYT. They combined high Zn genes from a number of genetic sources resulting in increases in grain Zn of 30-40% while retaining good agronomic performance¹⁷. A number of Zn biofortified wheat varieties have been released with grain yields comparable to those of conventional varieties and increases of 8–10 ppm (25–40%) of Zn in the grain¹⁸.”

- **Lines 330-332: Was there any overlap in the present study with the QTL identified in the studies cited in this sentence?**

The following was added: “Numerous QTLs and Marker-Trait Associations (MTAs) for Fe and Zn have been reported, based on Mendelian analysis and Genome Wide Association Studies (GWAS), with at least one QTL and/or MTA being reported on each of the 42 chromosomes^{26,27,31,33–37}. Velu et al. (2018) noted that QTLs for “nutrient uptake, transport and sequestration” are clustered on the group 2 and group 7 chromosomes and identified major QTL regions on 2B and 7B³³. We did not identify QTLs for Fe or Zn on either of these chromosomes. However, QTLs and/or MTAs have been reported on the same chromosomes as those reported here by other studies^{26,27,35,37}. It is possible that these corresponded to the QTL reported here but detailed comparisons of the genomic regions are required to confirm this.”

- **Lines 337-340, 346-353, 362-365, and parallel sentences throughout the discussion for other nutrients: Text regarding health implications of various mineral nutrients (and current fortification schema) seems better placed in the introduction as it is background information for this study.**

We think it is helpful to the reader to include the information here, rather than to need to refer back to the introduction. We have therefore retained the text in the discussion.

- **Lines 353-354: The increasing adoption of vegan diets is a cause of further concern and intakes from other foods need to be increased. It seems one or more references from the human nutrition realm would be helpful for this sentence.**

The following reference has been added: Bickelmann, F., Leitzmann, M., Keller, M., Baurecht, H. & Jochem, C. Calcium intake in vegan and vegetarian diets: A systematic review and Meta-analysis. Crit Rev Food Sci Nutr 63, 10659–10677 (2023).

- **Line 364: However, deficiency does occur in sheep and cattle, either due to grazing on pastures on low copper soils (without fertilisation) or due to ingestion of foods high in sulphur and molybdenum. Increasing the content of copper in feed grain could therefore be advantageous. Please lay out more explicitly for the reader why ingestion of foods high in sulphur and molybdenum is relevant for copper deficiency.**

This relates only to livestock and rather than provide details we have included a reference: Gooneratne, S. R. & Christensen, D. A. REVIEW OF COPPER DEFICIENCY AND METABOLISM IN RUMINANTS. Can J Anim Sci 69, 819–845 (1989).

- **Lines 368-369: It seems helpful to add ‘among other factors’ to the end of this sentence.**

This has been added.

- **Lines 385-387: This recent paper seems potentially relevant in the context of how physical compartments of the grain pertain to mineral nutrient concentrations (and whether biofortification efforts targeting physical traits could be efficacious):**
<https://academic.oup.com/g3journal/article/13/7/jkad085/7128031>

The paper describes increasing aleurone cell number and pericarp yield in maize. Studies have been carried out on increasing aleurone cell number in wheat, but this is not considered to be a realistic practical strategy for increasing mineral content due to effects on grain development and hence yield.

- **Line 394: It seems helpful to switch the order for Ca and Mg when referring to concentrations in white breadmaking flours. (Metrics are reported for Ca and then for Mg throughout the rest of the paragraph, but the order is switched in this last sentence in the paragraph.)**

This was done as suggested by the reviewer.

- **Lines 415-416: Were P and K levels kept constant when altering N levels? It would also be helpful to specify in what form the nitrogen (and fertilization more generally) was applied; formulation, timing and physical method of application, etc.**

These details are now provided in the Methods and in Supplementary Table 22.

- **Lines 420-421: Please report soil mineral nutrient results in the supplement.**

We only conducted soil N analysis. We did not determine other nutrients as it is difficult to accurately measure the concentration of available minerals in the soil. Furthermore, soils in the experimental sites are not considered to be deficient in micronutrients. Therefore, we did not include these measurements in our study.

- **Lines 426-7: More information on the protocol used for ICP-AES would be needed, including with reference to a published protocol if applicable. Related comment: Line 90 states 'ICP-OES', but lines 426-7 state ICP-AES; clarification would be needed.**

The following has been added to the Methods:

“Samples for mineral analysis were oven-dried at 80 °C overnight, weighed and digested using a mixture of nitric acid and perchloric acid (85:15 v/v) in open tube digestion blocks, followed by a programmed heating digestion: 60 °C for 180 minutes, 100 °C for 60 minutes, 120 °C for 60 minutes, 175 °C for 90 minutes and 50 °C until dry. The acids are removed by volatilisation and the residue dissolved in nitric acid (5% v/v). The elements were detected with Agilent 5900 SVDV Inductively Coupled Plasma - Optical Emission Spectrometer (ICP-OES) (Agilent Technologies LDA K Limited, Cheshire, UK). The analysis was strictly monitored using certified external standards alongside in-house standard materials. Standards and check samples are monitored and recorded using Shewhart Control Graphs and computer-based quality control packages.”

- **Line 427: Use of the phrase 'a known number' is unclear; were 1,000 grains counted or some subset thereof (and a multiplier then used to reach TGW)? Please specify.**

We have revised the manuscript to specify this, as follows: “TGW was determined by counting 1000 grains, drying at 105 °C overnight and recording the weight.”

- **Line 428: Were yield and TGW corrected/standardized for grain moisture? It would likely be helpful to specify either way. It seems this correction/standardization would have been feasible given the grain moisture data available from the NIRS instrument.**

In response to this comment, we have added details on the method followed for the grain and straw moisture content correction in the “Grain and Straw Analyses” subsection of the Materials and Methods.

- **Line 437: Which function within the R/qtl package was used for mapping? Were any non-default settings used? Please specify either way.**

As stated in the “Quantitative Genetics and Bioinformatics” section of the Materials & Methods, we used the R package ASMap (v1.0-4) for genetic mapping. We have oversight to report the p-value for defining linkage groups, many thanks for making us aware of this.

We have now updated the manuscript to include this information: “Genetic maps were constructed using package ASMap v1.0-452 following the same strategy as described in Min et al. (2020) using the p-values of 10-14 for Par x W160 and 10-12 for Par x W239 and Par x W292 to define linkage groups⁵³. The custom R code to conduct the mapping is deposited at: https://github.com/wingenl/genetic_mapping_with_ASMap.”

- **Line 458: What soil type (and mineral nutrition therein) were used in the glasshouse?**

More details were included in the corresponding Materials and Methods section.

- **Line 463: Are grain dimension traits (length and width) reported or discussed in this study? They seem potentially relevant, and it was not clear why they are not reported/discussed if they were measured for the TILLING lines and mentioned in the Materials and Methods section.**

These traits were measured and discussed only for the TILLING mutants. This is because mutant lines often exhibit shrunken grains, which could result in higher mineral concentrations due to reduced starch accumulation. The method used for determining the grain characteristics of the TILLING mutants is mentioned in the “Calcium candidate gene proof of function” subsection of the Materials and Methods section. Grain dimension measurement was not conducted for the RIL populations.

- **Lines 464-467: It would be helpful to report performance metrics for the calibration that was developed for the XRF instrument using data from the ICP-OES instrument; e.g., in cross-validation and on a held-out test set.**

The following is added to the methods:

“A cross-validation to the calibration was performed using additional 30 accessions. Linear regression correlation analysis was performed to compare ICP-OES data (ranging from 312 to 460 ppm) with XRF reads. The coefficient of determination, (R^2) was 0.823 and Pearson Correlation Coefficient value (R) was 0.907 with P value < 0.001 .”

- **Line 486: This reviewer would strongly encourage making the QTL analysis code publicly available at time of publication.**

The code used for the genetic mapping and QTL analysis became publicly available at GitHub.

We therefore updated the respective Materials and Methods section, as follows:

“The custom R code to conduct the mapping is deposited at: https://github.com/wingen/genetic_mapping_with_ASMap. QTL mapping was conducted using package qtl v1.5054 ,also as described in Min et al. (2020) using custom made scripts, available from https://github.com/wingen/rqtl_jic/.”

We also updated Code availability statements:

“The custom R code utilized for the genetic mapping and QTL analysis in this study is publicly available at GitHub. The code for genetic mapping can be accessed at https://github.com/wingen/genetic_mapping_with_ASMap. Similarly, the code for QTL analysis can be found at https://github.com/wingen/rqtl_jic/.”

- **Figure 2: It seems helpful to depict the centromere if not already.**

The centromeres are shown. In addition, the legend has also been revised as follows:

“Fig. 2: Distribution of identified QTLs for essential mineral content in bread wheat grain - This figure illustrates the approximate locations of the QTLs across the 21 chromosomes of wheat, based on the IWGSC RefSeq v1.0. Each colour annotation corresponds to a different mineral: Ca (Calcium), Cu (Copper), Fe (Iron), K (Potassium), Mg (Magnesium), and Zn (Zinc), as indicated by the legend. The scale beneath the chromosomes represents distances in Mb. Comprehensive details of the identified QTLs are provided in Table 2 and Supplementary Table 10.”

- **Table 1: It seems mean or median and standard deviation would be easier for readers to compare across populations, years, and nitrogen treatments than ranges.**

Mean values and SD were added in Table 1 along with the average concentration in parental lines for each population (Paragon and Watkins landraces).

- **A few points on Table 2:**

How was ‘major’ defined? It may be easiest to simply restate the LOD score threshold rather than introducing a new term.

“Major” was removed from Table legend. As reviewer suggested we restated our filtering criteria.

Adding 'W' in front of 292, 160, and 232 in the fourth column would be helpful in parsing this table.

This is done as suggested.

If the 'effects on straw and/or biomass' column is indicating whether a QTL was also detected in straw and/or biomass, it seems helpful to make the column header more clear/literal.

Column header was changed to "Straw QTL Co-location" to be more literal.

Is 'nearest' marker the same as peak marker? If so, it may be easiest to simply state 'peak marker' rather than introducing a new term.

This is revised as suggested.

Response to Reviewer #2 Comments

- **However, the population size of 94 F4 lines is too small, there is always a chance to identify too many spurious QTLs, use of 5 MB region for candidate genes listing is too big, there is no proper details on genotyping.**

Comments were covered in our response to general comments – please see above.

In relation to the spurious QTLs comment, all QTLs detected are statistically significant, so the number of spurious QTL should be low, within the statistical limits ($\alpha = 0.05$). Additionally, we only select QTLs that appear at least two different environments, which makes it much less likely that we have picked a spurious QTL.

- **Also impact of different levels of N on mineral accumulation/QTLs has not been done properly.**

Comments were covered in our response to general comments – please see above.

Only one cross was grown at two N levels with data for both levels for only two years. We do not think that these four (2 x N1, 2 x N2) samples sets are sufficient to carry out a detailed GxExM analysis. However, we provide the full datasets if the reader wishes to carry out this comparison.

- **Use of words such as strong and robust for QTLs needs clear justification or thresholds.**

This was changed and we instead used highest LOD score.

- **Induced mutagenesis for validation of gene for Ca does not make sense because Ca is not major target for biofortification.**

The following text has been added: The 5A Ca QTL had the highest LOD scores of all of the QTLs mapped in the study and was the most consistent, controlling Ca/grain and Ca concentration in a total of 8 sample sets from the two crosses with LOD scores ranging from 6.1 to 12.2 (Table 2). It was therefore selected for proof of function of candidate genes using mutant TILLING lines.

We accept that calcium is not a current target for biofortification, but its importance is demonstrated by the fact that white flours are fortified with Ca in some countries including the UK. Dietary sources of Ca should also be reviewed because of increasing risk of Ca deficiency in vegan diets (as discussed above).

- **Experiments have been well carried out and quite a lot of data has been generated but analyses, presentation and interpretation need a significant improvement. Data can be revisited with the help of biometrician. An in depth and updated analyses looking at the data for multiple aspects can provide a better information for the wheat biofortification.**

One of the authors is a professional bioinformatician. We appreciate that other aspects of the datasets can be explored, particularly in relation to processes in the plant. However, we think it is important to focus on the implications of the study for improving human health.

Response to Reviewer #3 Comments

- **The authors should add, also as supplementary material, the frequency distributions for the traits evaluated in the field trials.**

In response to your suggestion, we have added violin plots to illustrate the distribution of the concentration of the essential elements as Supplementary Fig. 2. These plots illustrate and compare the underlying distribution and summarize the data in each sample set.

- **How were the observed means considered statistically different? Based on the phenotypic distribution in the three populations, an ANOVA or a more suitable analysis should be carried out and shown.**

We reported the observed means but did not perform comparisons between group/population means etc, hence an ANOVA or similar statistical analysis was not conducted.

- **Table 1: add heritability and least significant difference to descriptive statistics.**

LSDs not reported as means are not compared. Heritability of the main trait of interest were added in Supplementary Table 23. The following was also added to the added to Materials and Methods under the paragraph: Quantitative Genetics and Bioinformatics. "The function 'repeatability' of

package heritability v1.4055 was used to calculate the broad sense heritability of the concentrations of minerals in grain and plant height, straw biomass, above ground biomass, grain yield, harvest index and thousand grain weight (Supplementary Table 23).”

- **Table 2: Add additive effect and R² for each major QTL.**

In response to this suggestion, we have now included the additive effect and the variance explained for each QTL in Table 2.

- **Add also the reference of papers in which the same region has been previously mapped if this is the case.**

The following text has been added:

“Numerous QTLs and Marker-Trait Associations (MTAs) for Fe and Zn have been reported, based on Mendelian analysis and Genome Wide association Studies (GWAS), with at least one QTL and/or MTA being reported on each of the 42 chromosomes (Krishnappa et al., 2017; 2021; 2022; Velu et al., 2018; Shariatipour et al., 2021; Tong et al., 2022; Wang et al., 2021; Juliana et al., 2022). Velu et al. (2018) noted that QTLs for “nutrient uptake, transport and sequestration” are clustered on the group 2 and group 7 chromosomes and identified major QTL regions on 2B and 7B. We did not identify QTLs for iron or zinc on either of these chromosomes. However, QTLs and/or MTAs have been reported on the same chromosomes as those reported here by other workers (Tong et al., 2022; Juliana et al., 2022; Krishnappa et al., 2017; 2022). It is possible that these corresponded to the QTL reported here but detailed comparisons of the genomic regions are required to confirm this.”

REVIEWERS' COMMENTS:

Reviewer #1 (Remarks to the Author):

The co-author team has fully and adeptly addressed my comments, which is appreciated. I happened to notice that 'carrplot' appears in the text (line 668 in the tracked-change version) rather than 'corrplot', in case the latter was meant and just in case helpful to point out.

Reviewer #2 (Remarks to the Author):

None

Reviewer #3 (Remarks to the Author):

It is not clear to me the reason why the Authors did not carry out any statistical analysis to compare the means shown in the study. I think this is an important aspect to be fixed.